# Spherical Motion Dynamics: Learning Dynamics of Neural Network with Normalization, Weight Decay, and SGD

## Abstract

In this work, we comprehensively reveal the learning dynamics of neural network with normalization, weight decay (WD), and SGD (with momentum), named as Spherical Motion Dynamics (SMD). Most related works study SMD by focusing on "effective learning rate" in "equilibrium" condition, i.e. assuming the convergence of weight norm. However, their discussions on why this equilibrium condition can be reached in SMD is either absent or less convincing. Our work investigates SMD by directly exploring the cause of equilibrium condition. Specifically, 1) we introduce the assumptions that can lead to equilibrium condition in SMD, and prove that weight norm can approach its theoretical value in a linear rate regime with given assumptions; 2) we propose "angular update" as a substitute for effective learning rate to measure the evolving of neural network in SMD, and prove angular update can also approach to its theoretical value in a linear rate regime; 3) we verify our assumptions and theoretical results on various computer vision tasks including ImageNet and MSCOCO with standard settings. Experiment results show our theoretical findings agree well with empirical observations.

## 1 Introduction and Background

Normalization techniques (e.g. Batch Normalization (Ioffe & Szegedy, 2015) or its variants) are one of the most commonly adopted techniques for training deep neural networks (DNN). A typical normalization can be formulated as following: consider a single unit in a neural network, the input is $\boldsymbol{X}$, the weight of linear layer is $\boldsymbol{w}$ (bias is included in $\boldsymbol{w}$), then its output is

$$y(\boldsymbol{X}; \boldsymbol{w}; \gamma; \beta) = g(\frac{\boldsymbol{X}\boldsymbol{w} - \mu(\boldsymbol{X}\boldsymbol{w})}{\sigma(\boldsymbol{w}\boldsymbol{X})}\gamma + \beta), \tag{1}$$

where $g$ is a nonlinear activation function like ReLU or sigmoid, $\mu$, $\sigma$ are mean and standard deviation computed across specific dimension of $\boldsymbol{X}\boldsymbol{w}$ (like Batch Normalization (Ioffe & Szegedy, 2015), Layer Normalization Ba et al. (2016), Group Normalization (Wu & He, 2018), etc.). $\beta$, $\gamma$ are learnable parameters to remedy for the limited range of normalized feature map. Aside from normalizing feature map, Salimans & Kingma (2016) normalizes weight by $l_2$ norm instead:

$$y(\boldsymbol{X}; \boldsymbol{w}; \gamma; \beta) = g(\boldsymbol{X}\frac{\boldsymbol{w}}{||\boldsymbol{w}||_2}\gamma + \beta), \tag{2}$$

where $|| \cdot ||_2$ denotes $l_2$ norm of a vector.

**Characterizing evolving of networks during training.** Though formulated in different manners, all normalization techniques mentioned above share an interesting property: the weight $\boldsymbol{w}$ affiliated with a normalized unit is *scale-invariant*: $\forall \alpha \in \mathbb{R}^+, y(\boldsymbol{X}; \alpha\boldsymbol{W}; \gamma, \beta) = y(\boldsymbol{X}; \boldsymbol{w}; \gamma, \beta)$. Due to the scale-invariant property of weight, the Euclidean distance defined in weight space completely fails to measure the evolving of DNN during learning process. As a result, original definition of learning rate $\eta$ cannot sufficiently represent the update efficiency of normalized DNN.

To deal with such issue, van Laarhoven (2017); Hoffer et al. (2018); Zhang et al. (2019) propose "effective learning rate" as a substitute for learning rate to measure the update efficiency of normalized

neural network with stochastic gradient descent (SGD), defined as

$$\eta_{eff} = \frac{\eta}{||\boldsymbol{w}||_2^2}. \tag{3}$$

**Joint effects of normalization and weight decay.** van Laarhoven (2017) explores the joint effect of normalization and weight decay (WD), and obtains the magnitudes of weight by assuming the convergence of weight, i.e. if $\boldsymbol{w}_t = \boldsymbol{w}_{t+1}$, the weight norm can be approximated as $||\boldsymbol{w}_t||_2 = O(\sqrt[4]{\eta/\lambda})$, where $\lambda$ is WD coefficient. Combining with Eq.(3), we have $\eta_{eff} = \sqrt{\eta\lambda}$. A more intuitive demonstration about relationship between normalization and weight decay is presented in Chiley et al. (2019) (see Figure 1): due to the fact that the gradient of scale invariant weight $\partial\mathcal{L}/\partial\boldsymbol{w}$ ($\mathcal{L}$ is the loss function of normalized network without WD part) is always perpendicular to weight $\boldsymbol{w}$, one can infer that gradient component $\partial\mathcal{L}/\partial\boldsymbol{w}$ always tends to increase the weight norm, while the gradient component provided by WD always tends to reduce weight norm. Thus if weight norm remains unchanged, or "equilibrium has been reached"[1], one can obtain

$$\frac{\boldsymbol{w}_t - \boldsymbol{w}_{t+1}}{||\boldsymbol{w}_t||_2} = \sqrt{2\eta\lambda}\frac{\partial\mathcal{L}/\partial\boldsymbol{w}}{\mathbb{E}||\partial\mathcal{L}/\partial\boldsymbol{w}||_2}. \tag{4}$$

Eq.(4) implies the magnitude of update is scale-invariant of gradients, and effective learning rate should be $\sqrt{2\eta\lambda}$. Li & Arora (2020) manages to estimate the magnitude of update in SGDM, their result is presented in limit and accumulation manner: if both $\lim_{T\longrightarrow\infty} R_T = 1/T\sum_{t=0}^{T}||\boldsymbol{w}_t||$ and $\lim_{T\longrightarrow\infty} D_T = 1/T\sum_{t=0}^{T}||\boldsymbol{w}_t - \boldsymbol{w}_{t+1}||$ exist, then we have

$$\lim_{T\longrightarrow\infty}\frac{D_T}{R_T} = \frac{2\eta\lambda}{1+\alpha}. \tag{5}$$

Though not rigorously, one can easily speculate from Eq.(5) the magnitude of update in SGDM cases should be $\sqrt{2\eta\lambda/(1+\alpha)}$ in equilibrium condition. But proof of Eq.(5) requires more strong assumptions: not only convergence of weight norm, but also convergence of update norm $||\boldsymbol{w}_{t+1} - \boldsymbol{w}_t||_2$ (both in accumulation manner).

As discussed above, all previous qualitative results about "effective learning rate" (van Laarhoven, 2017; Chiley et al., 2019; Li & Arora, 2020) highly rely on equilibrium condition, but none of them explores why such equilibrium condition can be achieved. Only van Laarhoven (2017) simply interprets the occurrence of equilibrium as a natural result of convergence of optimization, i.e. when optimization is close to be finished, $\boldsymbol{w}_t = \boldsymbol{w}_{t+1}$, resulting in equilibrium condition. However, this interpretation has an apparent contradiction: according to Eq.(4) and (5), when equilibrium condition is reached, the magnitude of update is constant, only determined by hyper-parameters, which means optimization process has not converged yet. Li & Arora (2020) also notices the non-convergence of SGD with BN and WD, so they do not discuss reasonableness of assumptions adopted by Eq.(5). In a word, previous results about "effective learning rate" in equilibrium condition can only provide vague insights, they are difficult to be connected with empirical observation.

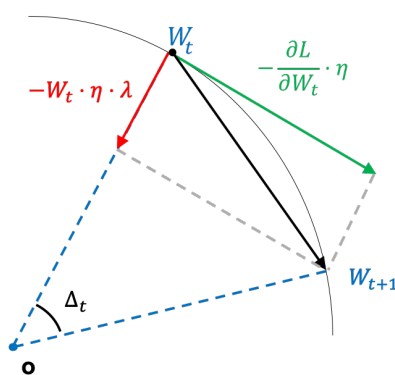

Figure 1: Illustration of optimization behavior with BN and WD. Angular update $\Delta_t$ represents the angle between the updated weight $\boldsymbol{W}_t$ and its former value $\boldsymbol{W}_{t+1}$.

In this work, we comprehensively reveal the learning dynamics of normalized neural network using stochastic gradient descent without/with momentum (SGD/SGDM) and weight decay, named as **Spherical Motion Dynamics (SMD)**. Our investigation aims to answer the following question:

---

[1]"Weight norm remains unchanged" means $||\boldsymbol{w}_t||_2 \approx ||\boldsymbol{w}_{t+1}||_2$, Chiley et al. (2019) calls this condition as "equilibrium", which will also be used in the following context of this paper. Note equilibrium condition is not mathematically rigorous, we only use it for intuitive analysis.

*Why and how equilibrium condition can be reached in Spherical Motion Dynamics?*

Specifically, our contributions are

- We introduce the assumptions that can lead to equilibrium condition in SMD, and justify their reasonableness by sufficient experiments. We also prove under given assumptions, equilibrium condition can be reached as weight norm approach to its theoretical value in a linear rate regime in SMD. Our assumptions show the equilibrium condition can occur long before the finish of the whole optimization;

- We define a novel index, *angular update*, to measure the change of normalized neural network in a single iteration, and derive its theoretical value under equilibrium condition in SMD. We also prove angular update can approach to its theoretical value in a linear regime along with behavior of weight norm. Our results imply that the update efficiency of SGD/SGDM on normalized neural network only depends on pre-defined hyper-parameters in both SGD and SGDM case;

- We verify our theorems on different computer vision tasks (including one of most challenging datasets ImageNet (Russakovsky et al., 2015) and MSCOCO (Lin et al., 2014)) with various networks structures and normalization techniques. Experiments show the theoretical value of angular update and weight norm agree well with empirical observation.

Recently, a parallel work (Li et al., 2020b) is published to analyze the equilibrium condition of normalized neural network with weight decay in SGD case. They model the learning dynamic of normalized network as a SDE, and propose the concept "intrinsic learning rate" ($\lambda_e = \lambda\eta$) to measure the converge rate of weight norm and effective learning rate. Their main result, in which converging time to equilibrium in SGD case is $\mathcal{O}(1/(\lambda\eta))$, is consistent with our theory. The major difference between Li et al. (2020b) and our work is that all results from Li et al. (2020b) are qualitatively results or conjectures limited on SGD case. While our work derives the quantitative results on both SGD/SGDM cases, our analysis allows us to precisely predict the value of weight norm/angular update in large scale data experiments, and clarify the difference of approaching rate between SGD and SGDM case. Besides, Li et al. (2020b) conducts empirical study to discuss the connection between the equilibrium state and accuracy of trained model. Our work mainly focuses on the equilibrium condition itself, connection between equilibrium condition and performance of trained model is beyond our discussion in this paper. The experiments we present in main text are only for verification of our theory.

Our theory on equilibrium condition in Spherical Motion Dynamics implies equilibrium condition mostly relies on the update rules of SGD/SGDM with WD, and scale-invariant property. The cause of equilibrium condition is independent of decrease of loss or trajectory of optimization, but equilibrium condition turns out to significantly affect update efficiency of normalized network by controlling the relative update (Eq.(4)). We believe equilibrium condition is one of the key reason why learning dynamic of normalized neural network is not consistent with traditional optimization theory (Li et al., 2020b). We think it has great potential to study the leaning dynamic of normalized network or develop novel efficient learning strategy under the view of Spherical Motion Dynamics.

## 2 RELATED WORK

**Normalization techniques** Batch normalization(BN (Ioffe & Szegedy, 2015)) is proposed to deal with gradient vanishing/explosion, and accelerate the training of DNN. Rapidly, BN has been widely used in almost all kinds of deep learning tasks. Aside from BN, more types of normalization techniques have been proposed to remedy the defects of BN (Ioffe, 2017; Wu & He, 2018; Chiley et al., 2019; Yan et al., 2020) or to achieve better performance (Ba et al., 2016; Ulyanov et al., 2016; Salimans & Kingma, 2016; Shao et al., 2019; Singh & Krishnan, 2020). Though extremely effective, the mechanism of BN still remains as a mystery. Existing works attempt to analyze the function of BN: Ioffe & Szegedy (2015) claims BN can reduce the Internal Covariance Shift (ICS) of DNN; Santurkar et al. (2018) argue that the effectiveness of BN is not related to ICS, but the smoothness of normalized network; Luo et al. (2019) shows BN can be viewed as an implicit regularization technique; Cai et al. (2019) proves that with BN orthogonal least square problem can converge at linear rate; Dukler et al. (2020) proves weight normalization can speed up training in a two-layer ReLU network.

**Weight decay** Weight decay (WD) is well-known as $l_2$ regularization, or ridge regression, in statistics. WD is also found to be extreme effective when applied in deep learning tasks. Krizhevsky & Geoffrey (2009) shows WD sometimes can even improve training accuracy not just generalization performance; Zhang et al. (2019) show WD can regularize the input-output Jacobian norm and reduce the effective damping coefficient; Li et al. (2020a) discusses the disharmony between WD and weight normalization. A more recent work Lewkowycz & Gur-Ari (2020) empirically finds the number of SGD steps $T$ until a model achieves maximum performance satisfies $T \propto \frac{1}{\lambda \eta}$, where $\lambda, \eta$ are weight decay factor and learning rate respectively, they interpret this phenomenon under the view of Neural Tangent Kernel (Jacot et al., 2018), showing that weight decay can accelerate the training process. Notice their result has no connection with equilibrium condition discussed in this work. Our results shows the cause of equilibrium condition can be reached long before neural network can get its highest performance.

**Effective learning rate** Due to the scale invariant property caused by normalization, researchers start to study the behavior of effective learning rate. van Laarhoven (2017); Chiley et al. (2019) estimate the magnitude of effective learning rate under equilibrium assumptions in SGD case; Hoffer et al. (2018) quantify effective learning rate without equilibrium assumptions, so their results are much weaker; Arora et al. (2019) proves that without WD, normalized DNN can still converge with fixed/decaying learning rate in GD/SGD cases respectively; Zhang et al. (2019) shows WD can increase effective learning rate; Li & Arora (2020) proves standard multi-stage learning rate schedule with BN and WD is equivalent to an exponential increasing learning rate schedule without WD. As a proposition, Li & Arora (2020) quantifies the magnitude of effective learning rate in SGDM case. But none of them have ever discussed why equilibrium condition can be reached. A recent work Li et al. (2020b) studies the convergence of effective learning rate by SDE, proving that the convergence time is of $\mathcal{O}(1/(\lambda \eta))$, where $\lambda, \eta$ are weight decay factor and learning rate respectively. Their result can only provide intuitive understanding, and is limited on SGD case.

## 3 PRELIMINARY ON SPHERICAL MOTION DYNAMICS

First of all, we review the property of scale invariant weight, and depict Spherical Motion Dynamics (SMD) in SGD case. Notice except definitions, all intuitive statements or derivations in this section mostly comes from previous literature, they are not mathematically rigorous. We summarize them to provide background of our topic and preliminary knowledge for readers.

**Lemma 1.** *If $w$ is scale-invariant with respect to $\mathcal{L}(w)$, then for all $k > 0$, we have:*

$$\langle \boldsymbol{w}_t, \frac{\partial \mathcal{L}}{\partial \boldsymbol{w}}\Big|_{\boldsymbol{w}=\boldsymbol{w}_t} \rangle = 0 \tag{6}$$

$$\frac{\partial \mathcal{L}}{\partial \boldsymbol{w}}\Big|_{\boldsymbol{w}=k\boldsymbol{w}_t} = \frac{1}{k} \cdot \frac{\partial \mathcal{L}}{\partial \boldsymbol{w}}\Big|_{\boldsymbol{w}=\boldsymbol{w}_t}. \tag{7}$$

Proof can be seen in Appendix B.1. Lemma 1 is also discussed in Hoffer et al. (2018); van Laarhoven (2017); Li & Arora (2020). Eq.(7) implies gradient norm is influenced by weight norm, but weight norm does not affect the output of DNN, thus we define **unit gradient** to eliminate the effect of weight norm.

**Definition 1** (Unit Gradient). *If $\boldsymbol{w}_t \neq \boldsymbol{0}$, $\tilde{\boldsymbol{w}} = \boldsymbol{w}/||\boldsymbol{w}||_2$, the unit gradient of $\partial \mathcal{L}/\partial \boldsymbol{w}|_{\boldsymbol{w}=\boldsymbol{w}_t}$ is $\partial \mathcal{L}/\partial \boldsymbol{w}|_{\boldsymbol{w}=\tilde{\boldsymbol{w}}_t}$.*

Setting $k$ as $1/||\boldsymbol{w}_t||_2$ inEq.(7), we have

$$\frac{\partial \mathcal{L}}{\partial \boldsymbol{w}}\Big|_{\boldsymbol{w}=\boldsymbol{w}_t} = \frac{1}{||\boldsymbol{w}_t||} \cdot \frac{\partial \mathcal{L}}{\partial \boldsymbol{w}}\Big|_{\boldsymbol{w}=\tilde{\boldsymbol{w}}_t}. \tag{8}$$

A typical SGD update rule without WD is

$$\boldsymbol{w}_{t+1} = \boldsymbol{w}_t - \eta \frac{\partial \mathcal{L}}{\partial \boldsymbol{w}}\Big|_{\boldsymbol{w}=\boldsymbol{w}_t}, \tag{9}$$

if $||\boldsymbol{w}_t||_2 = ||\boldsymbol{w}_{t+1}||_2$, dividing both side of Eq.(9) by $||\boldsymbol{w}_t||_2$, we have

$$\tilde{\boldsymbol{w}}_{t+1} = \tilde{\boldsymbol{w}}_t - \frac{\eta}{||\boldsymbol{w}_t||_2^2} \frac{\partial \mathcal{L}}{\partial \boldsymbol{w}}\Big|_{\boldsymbol{w}=\tilde{\boldsymbol{w}}_t} = \tilde{\boldsymbol{w}}_t - \eta_{eff} \cdot \frac{\partial \mathcal{L}}{\partial \boldsymbol{w}}\Big|_{\boldsymbol{w}=\tilde{\boldsymbol{w}}_t}. \tag{10}$$

Eq.(10) shows effective learning rate can be viewed as learning rate of SGD on unit sphere $\mathcal{S}^{p-1}$ (Hoffer et al., 2018). But effective learning rate still cannot properly represent the magnitude of update, since unit gradient norm is unknown. Therefore we propose the **angular update** defined below.

**Definition 2 (Angular Update).** *Assuming $\boldsymbol{w}_t$ is a scale-invariant weight from a neural network at iteration $t$, then the **angular update** $\Delta_t$ is defined as*

$$\Delta_t = \angle(\boldsymbol{w}_t, \boldsymbol{w}_{t+1}) = arccos\left(\frac{\langle \boldsymbol{w}_t, \boldsymbol{w}_{t+1} \rangle}{||\boldsymbol{w}_t|| \cdot ||\boldsymbol{w}_{t+1}||}\right), \tag{11}$$

*where $\angle(\cdot, \cdot)$ denotes the angle between two vectors, $\langle \cdot, \cdot \rangle$ denotes the inner product.*

According to Eq.(6), $\partial \mathcal{L}/\partial \boldsymbol{w}$ is perpendicular to weight $\boldsymbol{w}$. Therefore, if angular update $\Delta_t$ is small enough, it can be approximated by first order Taylor series expansion of $\tan \Delta_t$, then we can see its connection between effective learning rate and unit gradient norm

$$\Delta_t = \tan(\Delta_t) = \frac{\eta}{||\boldsymbol{w}_t||} \cdot ||\frac{\partial \mathcal{L}}{\partial \boldsymbol{w}}\Big|_{\boldsymbol{w}=\boldsymbol{w}_t}||_2 = \eta_{eff} \cdot \frac{\partial \mathcal{L}}{\partial \boldsymbol{w}}\Big|_{\boldsymbol{w}=\tilde{\boldsymbol{w}}_t}. \tag{12}$$

Another deduction from Eq.(6) is that weight norm always increases because

$$||\boldsymbol{w}_{t+1}||_2^2 = ||\boldsymbol{w}_t||_2^2 + (\eta||\frac{\partial \mathcal{L}}{\partial \boldsymbol{w}}\Big|_{\boldsymbol{w}=\boldsymbol{w}_t}||_2)^2 > ||\boldsymbol{w}_t||_2^2. \tag{13}$$

From Eq.(7) we can infer that increasing weight norm can lead to smaller gradient norm if unit gradient norm is unchanged. Zhang et al. (2019) states the potential risk that GD/SGD without WD but BN will converge to a stationary point not by reducing loss but by reducing effective learning rate due to increasing weight norm. Arora et al. (2019) proves that full gradient descent can avoid the risk and converge to a stationary point defined in $\mathcal{S}^{p-1}$, but their results still require sophisticated learning rate decay schedule in SGD case. Besides, practical implementation suggests training DNN without WD always suffers from poor generalization (Zhang et al., 2019; Bengio & LeCun, 2007; Lewkowycz & Gur-Ari, 2020).

Now considering the update rule of SGD with WD:

$$\boldsymbol{w}_{t+1} = \boldsymbol{w}_t - \eta\big(\frac{\partial \mathcal{L}}{\partial \boldsymbol{w}}\Big|_{\boldsymbol{w}=\boldsymbol{w}_t} + \lambda\boldsymbol{w}_t\big). \tag{14}$$

We can approximate the update of weight norm by

$$||\boldsymbol{w}_{t+1}||_2 \approx ||\boldsymbol{w}_t||_2 - \lambda\eta||\boldsymbol{w}_t||_2 + \frac{\eta^2}{2||\boldsymbol{w}_t||_2^3} \cdot ||\frac{\partial \mathcal{L}}{\partial \boldsymbol{w}}\Big|_{\boldsymbol{w}=\tilde{\boldsymbol{w}}_t}||_2^2. \tag{15}$$

The derivation of Eq.(15) is presented in Appendix A.1. Eq.(15) implies WD can provide direction to reduce weight norm, hence Chiley et al. (2019); Zhang et al. (2019) points out the possibility that weight norm can be steady, but do not explain this clearly. Here we demonstrate the mechanism deeper (see Figure 1): if unit gradient norm remains unchanged, note "centripetal force" $(-\lambda\eta||\boldsymbol{w}_t||_2)$ is proportional to weight norm, while "centrifugal force" $(\frac{\eta^2}{2||\boldsymbol{w}_t||_2^3} \cdot ||\frac{\partial \mathcal{L}}{\partial \boldsymbol{w}}\Big|_{\boldsymbol{w}=\tilde{\boldsymbol{w}}_t}||_2^2)$ is inversely proportional to cubic weight norm. As a result, the dynamics of weight norm is like a spherical motion in physics: overly large weight norm makes centripetal force larger than centrifugal force, leading to decreasing weight norm; while too small weight norm makes centripetal force smaller than centrifugal force, resulting in the increase of weight norm. Intuitively, equilibrium condition tends to be reached if the number of iterations is sufficiently large.

Notice that the core assumption mentioned above is "unit gradient norm is unchanged". In fact this assumption can solve the contradiction we present in Section 1: convergence of $\boldsymbol{w}_t$ leads to nonzero $||\boldsymbol{w}_{t+1} - \boldsymbol{w}_t||_2/||\boldsymbol{w}_t||_2$. Our analysis implies the convergence of weight norm is not equivalent to convergence of weight, steady unit gradient norm can also make weight norm converge, steady unit gradient norm does not rely on the fact that optimization has reached an optimum solution. But a problem about unit gradient assumption rises: unit gradient norm cannot remain unchanged during training in practice, it is not a reasonable assumption. In next section, we formulate this assumption in a reasonable manner, and rigorously prove the existence of equilibrium condition in SGD case. Discussion about SGD cannot be easily extended to SGDM case, because momentum is not always perpendicular to weight as unit gradient. But we can still prove the existence of equilibrium condition in SGDM case under modified assumptions.

## 4 MAIN RESULTS

First of all, we prove the existence of equilibrium condition in SGD case and provide the approaching rate of weight norm.

**Theorem 1.** *(Equilibrium condition in SGD) Assume the loss function is $\mathcal{L}(\boldsymbol{X}; \boldsymbol{w})$ with scale-invariant weight $\boldsymbol{w}$, denote $\boldsymbol{g}_t = \frac{\partial \mathcal{L}}{\partial \boldsymbol{w}}\big|_{\boldsymbol{X}_t, \boldsymbol{w}_t}$, $\tilde{\boldsymbol{g}}_t = \boldsymbol{g}_t \cdot ||\boldsymbol{w}_t||_2$. Considering the update rule of SGD with weight decay,*

$$\boldsymbol{w}_{t+1} = \boldsymbol{w}_t - \eta \cdot (\boldsymbol{g}_t + \lambda \boldsymbol{w}_t) \tag{16}$$

*where $\lambda, \eta \in (0, 1)$. If the following assumptions hold: 1) $\lambda\eta << 1$; 2) $\exists L, V, \in \mathbb{R}^+$, $\mathbb{E}[||\tilde{\boldsymbol{g}}_t||_2^2|\boldsymbol{w}_t] = L$, $\mathbb{E}[(||\tilde{\boldsymbol{g}}_t||_2^2 - L)^2|\boldsymbol{w}_t] \le V$; 3) $\exists l \in \mathbb{R}^+$, $||\tilde{\boldsymbol{g}}_t||_2^2 > l$, $l > 2[\frac{2\lambda\eta}{1-2\lambda\eta}]^2 L$. Then $\exists B > 0$, $w^* = \sqrt[4]{L\eta/(2\lambda)}$, we have*

$$\mathbb{E}[||\boldsymbol{w}_T||_2^2 - (w^*)^2]^2 \le (1 - 4\lambda\eta)^T B + \frac{V\eta^2}{l}. \tag{17}$$

**Remark 1.** *The theoretical value of weight norm $w^*$ in Theorem 1 is consistent with the derivation of weight norm in equilibrium in van Laarhoven (2017), though van Laarhoven (2017) assumes the equilibrium condition has been reached in advance, hence van Laarhoven (2017) cannot provide the approaching rate and scale of bias/variance. The vanishing term $((1 - 4\lambda\eta)^T B)$ in Eq.(17) is consistent with the convergence time $\mathcal{O}(1/(\lambda\eta))$ presented in Li et al. (2020b).*

Proof of Theorem 1 can be seen in Appendix B.2. It shows the square of weight norm can approach to its theoretical value in a linear rate regime (when vanishing term is larger than noisy term in Eq.(17)), and its variance is bounded by $\frac{V\eta^2}{l}$, which is empirically small. Now we discuss the reasonableness of assumptions in Theorem 1. Assumption 1 is consistent with commonly used settings; assumptions 2 and 3 imply the $\mathbb{E}||\tilde{g}_t||_2^2$ remains unchanged within several iterations, and its lower bound cannot be far from its expectation.

We need to clarify assumption 2 further: we do not require the $\mathbb{E}||\tilde{g}_t||_2^2$ must be constant across the whole training process, but only remains unchanged locally. On one hand, small learning rate($\eta$) can guarantee $\mathbb{E}||\tilde{g}_t||_2^2$ changes slowly; on the other hand, when $\mathbb{E}||\tilde{g}_t||_2^2$ changes, it means that square of weight norm will approach to its new theoretical value as Theorem 1 describes. Experiment result in Figure 2 strongly justifies our analysis. We also conduct extensive experiments to verify our claim further, please refer to Appendix C.1.

Now we extend Theorem 1 to SGDM case. SGDM is more complex than SGD since momentum is not always perpendicular to the weight, therefore we need to modify assumptions.

**Theorem 2.** *(Equilibrium condition in SGDM) Considering the update rule of SGDM (heavy ball method (Polyak, 1964)):*

$$\boldsymbol{v}_t = \alpha\boldsymbol{v}_{t-1} + \boldsymbol{g}_t + \lambda\boldsymbol{w}_t \tag{18}$$
$$\boldsymbol{w}_{t+1} = \boldsymbol{w}_t - \eta\boldsymbol{v}_t \tag{19}$$

*where $\lambda, \eta, \alpha \in (0, 1)$. If following assumptions hold: 4) $\lambda\eta << 1$, $\lambda\eta < (1 - \sqrt{\alpha})^2$; 5) Define $h_t = ||\boldsymbol{g}_t||_2^2 + 2\alpha\langle\boldsymbol{v}_{t-1}, \boldsymbol{g}_t\rangle$, $\tilde{h}_t = h_t \cdot ||\boldsymbol{w}_t||_2^2$. $\exists L, V, \in \mathbb{R}^+$, $\mathbb{E}[\tilde{h}_t|\boldsymbol{w}_t] = L$, $\mathbb{E}[(\tilde{h}_t - L)^2|\boldsymbol{w}_t] \le V$; 6)$\exists l \in \mathbb{R}^+$, $h_t > l$, $l > 2[\frac{6\lambda\eta}{(1-\alpha)^3(1+\alpha)-8\lambda\eta(1-\alpha)}]^2 L$. then $\exists B > 0$, $w^* = \sqrt[4]{L\eta/(\lambda(1-\alpha)(2 - \lambda\eta/(1+\alpha)))}$, we have*

$$\mathbb{E}[||\boldsymbol{w}_T||_2^2 - (w^*)^2]^2 \le 3B \cdot (1 - \frac{4\lambda\eta}{1-\alpha})^T + \frac{3V\eta^2(1 + 4\alpha^2 + \alpha^4)}{l(1-\alpha)^4}, \tag{20}$$

**Remark 2.** *So far, no other work rigorously prove equilibrium condition can be reached in SGDM cases. Even the most relevant work (Li et al., 2020b) only provides their conjecture on approaching rate of weight norm in SGDM, they speculate that the time of approaching to equilibrium should be $\mathcal{O}(1/(\lambda\eta))$, same order as approaching time in SGD case, their conjecture cannot provide further insight. While our results (vanishing terms in Eq.(17), (20) respectively) can clearly reflect the difference: the approaching rate of SGDM should be $1/(1 - \alpha)$ times faster than rate of SGD with same $\eta\lambda$. $\alpha$ is usually set as $0.9$, so SGDM can reach equilibrium condition much faster than SGD.*

Proof can be seen in Appendix B.3. Like assumption 1, assumption 4 is also satisfied for commonly used hyper-parameter settings. Besides, $\lambda\eta < (1 - \sqrt{\alpha})^2$ is also mentioned in Li & Arora (2020) for other purpose; Assumption 5 shows not unit gradient gradient norm $||\tilde{\boldsymbol{g}}_t||_2^2$ but an adjusted value $\tilde{h}_t$ dominates the expectation and variance of the weight norm square. We empirically find the expectation of $\langle \boldsymbol{v}_{t-1}, g_t \rangle$ is very close to 0, therefore the behavior of $\tilde{h}_t$ is similar to that of $||\tilde{\boldsymbol{g}}_t||_2^2$ (see (d) in Figure 2), making square of weight norm approaching its theoretical value in SGDM case. We leave theoretical analysis on $\tilde{h}_t$ as future work. As for assumption 6, commonly used settings ($\eta\lambda \ll 1$) can make $2[\frac{6\lambda\eta}{(1-\alpha)^3(1+\alpha)-8\lambda\eta(1-\alpha)}]^2$ as an very small lower bound for $l/L$. The experiments on justification of assumptions 4,5,6 can be seen in Figure 2 and appendix C.1. Comparing with Eq.(17) and Eq.(20), it implies that with same $\eta, \lambda$, SGDM can reach equilibrium condition much faster than SGD, but may have a larger variance, our experiments also verify that(see (b), (e) in Figure 2).

Since we have proven weight norm will approach to its theoretical value in SMD in a linear rate regime, we can derive the theoretical value of angular update $\Delta_t$ and its variance.

**Theorem 3.** *(Theoretical value of Angular Update) In SGD case, if assumptions in theorem 1 holds, then $\exists C > 0$, we have*

$$\mathbb{E}(\Delta_T - \sqrt{2\lambda\eta})^2 < (1 - 4\eta\lambda)^T C + \frac{V}{Ll} \tag{21}$$

*In SGDM case, if assumptions in theorem 2 holds, $\exists C > 0$, we have*

$$\mathbb{E}(\Delta_t - \sqrt{\frac{2\lambda\eta}{1+\alpha}})^2 < (1 - \frac{4\eta\lambda}{1-\alpha})^T C + \frac{(1-\alpha^2)(1+4\alpha^2+\alpha^4)V}{4Ll\alpha^4} \tag{22}$$

**Remark 3.** *Theoretical value of angular update in Theorem 3 is consistent with Eq.(4, 5) from Chiley et al. (2019); Li & Arora (2020) respectively. Notice the variance term in Eq.(21,22) is of $O(V/Ll)$, it is too large comparing with its empirical value, we leave it as a future work to improve the bound of variance term. Though connection between performance of neural network and equilibrium condition is beyond the main discussion of this paper, the findings of a parallel work (Li et al., 2020b) can inspire us the possible advantage of momentum: Li et al. (2020b) interprets that smaller $\lambda\eta$ can get higher performance when training DNN, but the approaching rate ($O(\lambda\eta)$) is slow; larger $\lambda\eta$ has faster approaching rate but it can only get bad performance. Inspired by Li et al. (2020b), we suspect that $\Delta_t \propto \sqrt{\lambda\eta}$ is highly correlated to the performance of DNN: smaller angular update in equilibrium condition can lead to better performance of DNN. According to theorem 3, with same $\lambda\eta$, momentum method has faster approaching rate($\lambda\eta/(1-\alpha)$) and smaller angular update ($\sqrt{2\lambda\eta/(1+\alpha)}$) than pure SGD. This means momentum method can simultaneously accelerate training process and improve the final performance, comparing pure SGD method. We leave this conjecture as a future work.*

Proof of theorem 3 is shown in Appendix B.4. According to theorem 3, the theoretical value of angular update and its approaching rate almost only depends on hyper-parameters: weight decay factor $\lambda$, learning rate $\eta$, (and momentum factor $\alpha$). It implies update efficiency of scale-invariant weights within a single step is totally controlled by predefined hyper-parameters in equilibrium condition, regardless other attributes of the weights (shape, size, position in network structure, or effects from other weights). That's the key reasons why we propose Angular Update (Eq.(11)) to replace effective learning rate (Eq.(3)): effective learning rate can only reflect the influence of weight norm, it cannot reveal how unit gradient norm affect the relative update (see Eq.(10)), while angular update in equilibrium condition implies that both weight norm and unit gradient norm do not affect the scale of update within a single iteration, only hyper-parameters (learning rate, weight decay factor, momentum factor) matter. Our experiment results strongly prove our claim (see Figure 3(a), 3(b), 3(d), 3(e)).

## 5 EXPERIMENTS

In this section, we show that our theorems can agree well with empirical observations on ImageNet (Russakovsky et al., 2015) and MSCOCO (Lin et al., 2014). We conduct experiments in two cases. In the first case we train neural network with fixed learning rate to verify our theorems in SGD and SGDM, respectively; in the second case we investigate SMD with more commonly used settings, multi-stage learning rate schedule.

## 5.1 FIXED LEARNING RATE

With fixed learning rate, we train Resnet50 (He et al., 2016) with SGD/SGDM on ImageNet. Learning rate is fixed as $\eta = 0.2$; WD factor is $\lambda = 10^{-4}$; with SGDM, the momentum factor is $\alpha = 0.9$. Figure 2 presents the unit gradient norm square, weight norm, and angular update of the weights from LAYER.2.0.CONV2 of Resnet50 in SGD and SGDM cases, respectively. It can be inferred from Figure 2 the behavior of $||\tilde{\boldsymbol{g}}_t||_2^2$, $\tilde{h}_t$ and hyper-parameter settings satisfy our assumptions in Theorem 1 and 2, therefore theoretical value of weight norm and angular update agree with empirical value very well. We also can observe SGDM can achieve equilibrium more quickly than SGD. According to Eq.(17),(20), the underlying reason might be with same learning rate $\eta$ and WD factor $\lambda$, approaching rate of SGDM $(1 - \frac{\lambda\eta}{1-\alpha})$ is smaller than that of SGD $(1 - \lambda\eta)$.

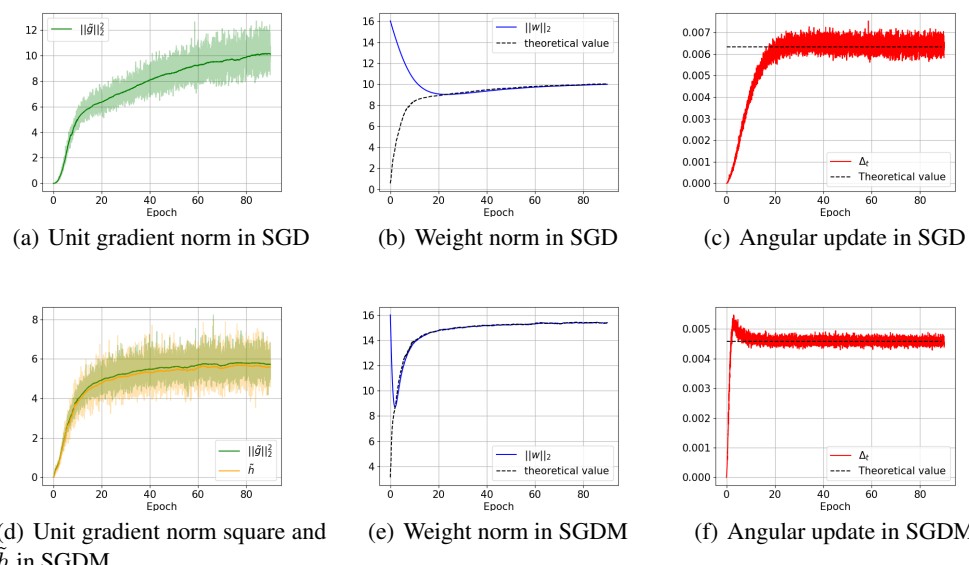

(a) Unit gradient norm in SGD   (b) Weight norm in SGD   (c) Angular update in SGD

(d) Unit gradient norm square and $\tilde{h}$ in SGDM   (e) Weight norm in SGDM   (f) Angular update in SGDM

Figure 2: Performance of *layer.2.0.conv2* from Resnet50 in SGD and SGDM, respectively. In (a), (d), semitransparent line represents the raw value of $||\tilde{\boldsymbol{g}}_t||_2^2$ or $\tilde{h}_t$, while solid line represents the averaged value within consecutive 200 iterations to estimate the expectation of $||\tilde{\boldsymbol{g}}_t||_2^2$ or $\tilde{h}_t$; In (b), (e), blue solid lines shows the raw value of weight norm $||\boldsymbol{w}_t||_2$, while dashed line represent the theoretical value of weight norm computed in Theorem 1, 2 respectively. The expectations $\mathbb{E}||\tilde{\boldsymbol{g}}||_2^2$ and $\mathbb{E}\tilde{h}$ are estimated as solid lines (a) and (d) respectively; In (c), (f), red lines represent raw value of angular update during training, dashed lines represent the theoretical value of angular update computed by $\sqrt{2\lambda\eta}$ and $\sqrt{2\lambda\eta/(1+\alpha)}$ respectively.

## 5.2 MULTI-STAGE LEARNING RATE SCHEDULE

Now we turn to study the behavior of angular update with SGDM and multi-stage leanring rate schedule on Imagenet (Russakovsky et al., 2015) and MSCOCO (Lin et al., 2014). In ImageNet classification task, we still adopt Resnet50 as baseline for it is a widely recognized network structure.The training settings rigorously follow Goyal et al. (2017): learning rate is initialized as 0.1, and divided by 10 at 30, 60, 80-th epoch; the WD factor is $10^{-4}$; the momentum factor is 0.9. In MSCOCO experiment, we conduct experiments on Mask-RCNN (He et al., 2017) benchmark using a Feature Pyramid Network (FPN) (Lin et al., 2017), ResNet50 backbone and SyncBN (Peng et al., 2018) following the 4x setting in He et al. (2019): total number of iteration is $360,000$, learning rate is initialized as 0.02, and divided by 10 at iteration $300,000$, $340,000$; WD coefficient is $10^{-4}$.

There appears to be some mismatch between theorems and empirical observations in (a), (b) of Figure 3: angular update in the last two stages is smaller than its theoretical value. This phenomenon can be well interpreted by our theory: according to Theorem 1, 2, when equilibrium condition is reached, theoretical value of weight norm satisfies $||\boldsymbol{w}_t||_2 \propto \sqrt[4]{\frac{\eta}{\lambda}}$, therefore when learning rate is

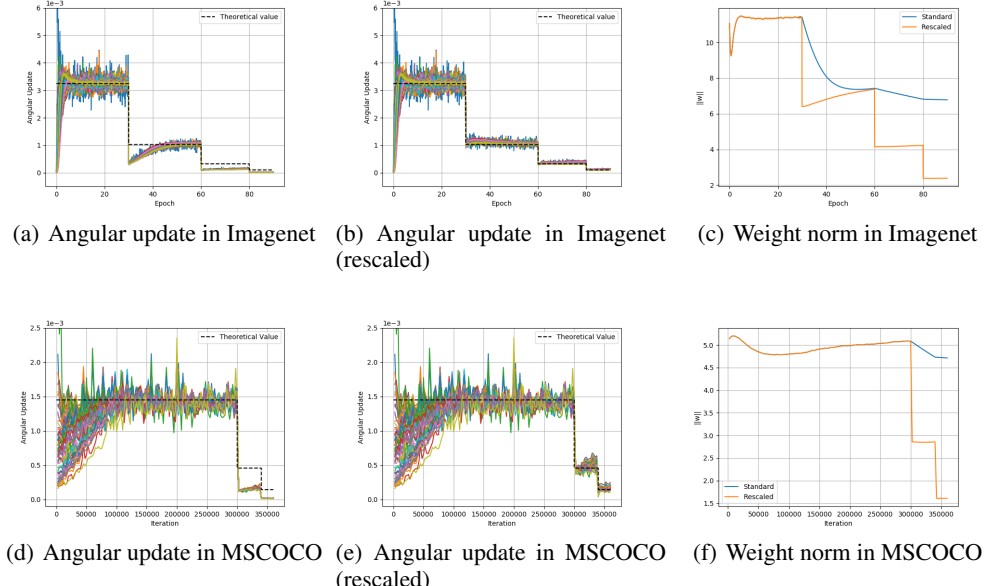

(a) Angular update in Imagenet   (b) Angular update in Imagenet (rescaled)   (c) Weight norm in Imagenet

(d) Angular update in MSCOCO   (e) Angular update in MSCOCO (rescaled)   (f) Weight norm in MSCOCO

Figure 3: In (a),(b),(d),(e), solid lines with different colors represent raw value of angular update of weights from all convolutional layer in the model; In (a), (d), training setting rigorously follows Goyal et al. (2017); He et al. (2019) respectively; In (b), (e), weight norm is divivied by $\sqrt[4]{10}$ as long as learning rate is divided by 10; In (c), (f), weight norm is computed on *layer.1.0.conv2* in Resnet50 backbone. Blue line represent original settings, orange line represent rescaled settings.

divided by $k$, equilibrium condition is broken, theoretical value of weight norm in new equilibrium condition should get $\sqrt[4]{1/k}$ smaller. But new equilibrium condition cannot be reached immediately (see (c), (f) in Figure 3), following corollary gives the least number of iterations to reach the new equilibrium condition.

**Corollary 3.1.** *In SGD case with learning rate $\eta$, WD coefficient $\lambda$, if learning rate is divided by $k$, and unit gradient norm remains unchanged, then at least $\lceil \lceil \log(k) \rceil / (4\lambda\eta) \rceil$ iterations are required to reach the new equilibrium condition; In SGDM case with momentum coefficient $\alpha$, then at least $\lceil \lceil \log(k)(1-\alpha) \rceil / (4\lambda\eta) \rceil$ iterations are required to reach the new equilibrium condition.*

Corollary 3.1 also implies SGD/SGDM with smaller learning rate requires more iterations to reach new equilibrium condition. Hence, in second learning rate stage in Imagenet experiments, angular update can reach its new theoretical value within 15 epochs, but during last two learning rate stages of Imagenet/MSCOCO experiments, SGDM cannot completely reach new equilibrium by the end of training procedure. As a result, we observe empirical value of angular update seems smaller than its theoretical value. Based on our theorem, we can bridge the gap by skipping the intermediate process from old equilibrium to new one. Specifically, when learning rate is divided by $k$, norm of scale-invariant weight is also divided by $\sqrt[4]{k}$, SGDM can reach in new equilibrium immediately. Experiments((b),(e) in Figure 3) show this simple strategy can make angular update always approximately equal to its theoretical value across the whole training process though learning rate changes.

## 6    CONCLUSION

In this paper, we comprehensively reveal the learning dynamics of DNN with normalization, WD, and SGD/SGDM, named as Spherical Motion Dynamics (SMD). Different from most related works (van Laarhoven, 2017; Hoffer et al., 2018; Chiley et al., 2019; Li & Arora, 2020), we directly explore the cause of equilibrium. Specifically, we introduce the assumptions that can lead to equilibrium condition, and show these assumptions are easily satisfied by practical implementation of DNN; Under given assumptions, we prove equilibrium condition can be reached at linear rate, far before optimization has converged. Most importantly, we show our theorem is widely valid, they can be verified on one of most challenging computer vision tasks, beyond synthetic datasets. We believe our theorems on SMD can bridge the gap between current theoretical progress and practical usage on deep learning techniques.

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

## A    MISSING DERIVATIONS

### A.1    APPROXIMATION OF WEIGHT NORM UPDATE

Now considering the update rule of SGD with WD:

$$\boldsymbol{w}_{t+1} = \boldsymbol{w}_t - \eta\left(\frac{\partial\mathcal{L}}{\partial\boldsymbol{w}}\Big|_{\boldsymbol{w}=\boldsymbol{w}_t} + \lambda\boldsymbol{w}_t\right). \tag{23}$$

Since $\frac{\partial\mathcal{L}}{\partial\boldsymbol{w}}\Big|_{\boldsymbol{w}=\boldsymbol{w}_t}$ is perpendicular to $\boldsymbol{w}_t$, we have

$$
\begin{aligned}
||\boldsymbol{w}_{t+1}||_2^2 &= ||\boldsymbol{w}_t - \eta\left(\frac{\partial\mathcal{L}}{\partial\boldsymbol{w}}\Big|_{\boldsymbol{w}=\boldsymbol{w}_t} + \lambda\boldsymbol{w}_t\right)||_2^2 & (24)\\
&= (1-\lambda\eta)^2||\boldsymbol{w}_t||_2^2 + \eta^2||\frac{\partial\mathcal{L}}{\partial\boldsymbol{w}}\Big|_{\boldsymbol{w}=\boldsymbol{w}_t}||_2^2. & (25)
\end{aligned}
$$

Therefore

$$
\begin{aligned}
||\boldsymbol{w}_{t+1}||_2 - ||\boldsymbol{w}_t||_2 &= \sqrt{(1-\lambda\eta)^2||\boldsymbol{w}_t||_2^2 + \eta^2||\frac{\partial\mathcal{L}}{\partial\boldsymbol{w}}\Big|_{\boldsymbol{w}=\boldsymbol{w}_t}||_2^2} - ||\boldsymbol{w}_t||_2 & (26)\\
&= \frac{(1-\lambda\eta)^2||\boldsymbol{w}_t||_2^2 + \eta^2||\frac{\partial\mathcal{L}}{\partial\boldsymbol{w}}\Big|_{\boldsymbol{w}=\boldsymbol{w}_t}||_2^2 - ||\boldsymbol{w}_t||_2^2}{\sqrt{(1-\lambda\eta)^2||\boldsymbol{w}_t||_2^2 + \eta^2||\frac{\partial\mathcal{L}}{\partial\boldsymbol{w}}\Big|_{\boldsymbol{w}=\boldsymbol{w}_t}||_2^2} + ||\boldsymbol{w}_t||_2}. & (27)
\end{aligned}
$$

Now assume $\eta, \lambda$ is extremely small, so that

$$(1-\lambda\eta)^2||\boldsymbol{w}_t||_2^2 + \eta^2||\frac{\partial\mathcal{L}}{\partial\boldsymbol{w}}\Big|_{\boldsymbol{w}=\boldsymbol{w}_t}||_2^2 - ||\boldsymbol{w}_t||_2^2 \approx -2\lambda\eta||\boldsymbol{w}_t||_2^2 + \eta^2||\frac{\partial\mathcal{L}}{\partial\boldsymbol{w}}\Big|_{\boldsymbol{w}=\boldsymbol{w}_t}||_2^2 \tag{28}$$

$$\sqrt{(1-\lambda\eta)^2||\boldsymbol{w}_t||_2^2 + \eta^2||\frac{\partial\mathcal{L}}{\partial\boldsymbol{w}}\Big|_{\boldsymbol{w}=\boldsymbol{w}_t}||_2^2} + ||\boldsymbol{w}_t||_2 \approx 2||\boldsymbol{w}_t||_2. \tag{29}$$

Therefore, we have

$$||\boldsymbol{w}_{t+1}||_2 - ||\boldsymbol{w}_t||_2 \approx -\lambda\eta||\boldsymbol{w}_t||_2 + \frac{\eta^2}{2||\boldsymbol{w}_t||_2}||\frac{\partial\mathcal{L}}{\partial\boldsymbol{w}}\Big|_{\boldsymbol{w}=\boldsymbol{w}_t}||_2^2 = -\lambda\eta||\boldsymbol{w}_t||_2 + \frac{\eta^2}{2||\boldsymbol{w}_t||_2^3}||\frac{\partial\mathcal{L}}{\partial\boldsymbol{w}}\Big|_{\boldsymbol{w}=\tilde{\boldsymbol{w}}_t}||_2^2 \tag{30}$$

where $\frac{\partial\mathcal{L}}{\partial\boldsymbol{w}}\Big|_{\boldsymbol{w}=\tilde{\boldsymbol{w}}_t}$ is the unit gradient defined in Definition 1.

## B    PROOF OF THEOREMS

**Remark 4.** *In the following context, we will use the following conclusion multiple times: $\forall\delta,\varepsilon\in\mathbb{R}$, if $|\delta|\ll 1, |\varepsilon|\ll 1$, then we have:*

$$(1+\delta)^2 \approx 1+2\delta, \quad \sqrt{1+\delta}\approx 1+\frac{\delta}{2}, \quad \frac{1}{1+\delta}\approx 1-\delta, \quad (1+\delta)(1+\varepsilon)\approx 1+\delta+\varepsilon. \tag{31}$$

### B.1    PROOF OF LEMMA 1

*Proof.* Given $\boldsymbol{w}_0 \in \mathbb{R}^p\backslash\{\boldsymbol{0}\}$, since $\forall k > 0, \mathcal{L}(\boldsymbol{w}_0) = \mathcal{L}(k\boldsymbol{w}_0)$, then we have

$$
\begin{aligned}
\frac{\partial\mathcal{L}(\boldsymbol{w})}{\partial\boldsymbol{w}}\Big|_{\boldsymbol{w}=\boldsymbol{w}_0} &= \frac{\partial\mathcal{L}(k\boldsymbol{w})}{\partial\boldsymbol{w}}\Big|_{\boldsymbol{w}=\boldsymbol{w}_0} = \frac{\partial\mathcal{L}(\boldsymbol{w})}{\partial\boldsymbol{w}}\Big|_{\boldsymbol{w}=k\boldsymbol{w}_0}\cdot k & (32)\\
\frac{\partial\mathcal{L}(k\boldsymbol{w})}{\partial k}\Big|_{\boldsymbol{w}=\boldsymbol{w}_0} &= \langle\frac{\partial\mathcal{L}(\boldsymbol{w})}{\partial\boldsymbol{w}}\Big|_{\boldsymbol{w}=k\boldsymbol{w}_0}, \boldsymbol{w}_0\rangle = \frac{1}{k}\cdot\langle\frac{\partial\mathcal{L}(\boldsymbol{w})}{\partial\boldsymbol{w}}\Big|_{\boldsymbol{w}=\boldsymbol{w}_0}, \boldsymbol{w}_0\rangle = 0 & (33)
\end{aligned}
$$

$\square$

### B.2 PROOF OF THEOREM 1

**Lemma 2.** *If the sequence $\{x_t\}_{t=1}^{\infty}$ satisfies*

$$x_t \geq \alpha x_{t-1} + \frac{L}{x_{t-1}} \tag{34}$$

*. where $x_1 > 0$, $L > 0$*

Then, we have

$$x_t \geq \sqrt{\frac{L}{1-\alpha}} - \alpha^{t-1} |\sqrt{\frac{L}{1-\alpha}} - x_1| \tag{35}$$

*Proof.* If $x_t \geq \sqrt{\frac{L}{1-\alpha}}$, since $\sqrt{L/(1-\alpha)} \geq 2\sqrt{\alpha L}$ then we have

$$x_{t+1} \geq \alpha x_t + \frac{L}{x_t} \geq \alpha \sqrt{\frac{L}{1-\alpha}} + \frac{L}{\sqrt{L/(1-\alpha)}} = \sqrt{\frac{L}{1-\alpha}}, \tag{36}$$

which means $\forall k > t, x_k \geq \sqrt{L/(1-\alpha)}$. If $x_t < \sqrt{L/(1-\alpha)}$, then we have

$$\sqrt{\frac{L}{1-\alpha}} - x_{t+1} \leq (\alpha - \frac{\sqrt{L(1-\alpha)}}{x_t})(\sqrt{\frac{L}{1-\alpha}} - x_t) < \alpha(\sqrt{\frac{L}{1-\alpha}} - x_t). \tag{37}$$

Therefore, by deduction method we have if $x_T < \sqrt{L/(1-\alpha)}$, then $\forall t \in [1, T-1]$, we have

$$0 < x_t < x_{t+1} \quad < \quad X_T < \sqrt{\frac{L}{1-\alpha}}, \tag{38}$$

$$(\sqrt{\frac{L}{1-\alpha}} - x_t) \quad < \quad \alpha^{t-1}(\sqrt{\frac{L}{1-\alpha}} - x_1). \tag{39}$$

In summary, we have

$$x_t \geq \sqrt{\frac{L}{1-\alpha}} - \alpha^{t-1} |\sqrt{\frac{L}{1-\alpha}} - x_1| \tag{40}$$

$\square$

*Proof of Theorem 1.* since $\langle \boldsymbol{w}_t, \boldsymbol{g}_t \rangle = 0$, then we have:

$$||\boldsymbol{w}_{t+1}||_2^2 = (1-\eta\lambda)^2 ||\boldsymbol{w}_t||_2^2 + \frac{||\tilde{\boldsymbol{g}}_t||_2^2 \eta^2}{||\boldsymbol{w}_t||_2^2} \tag{41}$$

Denote $x_t$ as $||\boldsymbol{w}_t||_2^2$, $L_t$ as $||\tilde{\boldsymbol{g}}_t||_2^2$ and omit $O((\eta\lambda)^2)$ part. Then $L_t > l$, $\mathbb{E}[L_t|x_t] = L$, $Var(L_t|x_t) = \mathbb{E}[(L_t - L)^2|x_t] < V$. Eq.(41) is equivalent to

$$x_{t+1} = (1 - 2\lambda\eta)x_t + \frac{L_t\eta^2}{x_t}. \tag{42}$$

According to Lemma 2, we have

$$x_t > \sqrt{\frac{l\eta}{2\lambda}} - (1 - 2\lambda\eta)^{t-1} |x_0 - \sqrt{\frac{l\eta}{2\lambda}}|. \tag{43}$$

Eq.(43) implies when $t > 1 + \frac{log((\sqrt{2}-1)\sqrt{l\eta/(4\lambda)}) - log(|x_0 - \sqrt{l\eta/(4\lambda)}|)}{log(1-2\lambda\eta)}$, we have $x_t > \sqrt{\frac{l\eta}{4\lambda}}$.

Now, denote $x^*$ as the $\sqrt{\frac{L\eta}{2\lambda}}$, then we have

$$\mathbb{E}[(x_{t+1} - x^*)^2|x_t] = \mathbb{E}[((1 - 2\lambda\eta - \frac{L\eta^2}{x_t x^*})(x_t - x^*) + \frac{L_t - L}{x_t})^2|x_t]$$

$$= (1 - 2\lambda\eta - \frac{L\eta^2}{x_t x^*})^2 (x_t - x^*)^2 + \frac{\mathbb{E}[(L_t - L)^2|x_t]\eta^4}{x_t^2} \tag{44}$$

If $t > T(\alpha, \lambda, \eta, l, x_0) = 1 + \frac{log((\sqrt{2}-1)\sqrt{l\eta/(4\lambda)}) - log(|x_0 - \sqrt{l\eta/(4\lambda)}|)}{log(1 - 2\lambda\eta)}$, Eq.(43) implies

$$1 - 2\lambda\eta - \sqrt{\frac{2L}{l}} \cdot 2\lambda\eta < 1 - 2\lambda\eta - \frac{L\eta^2}{x^* x_t}. \tag{45}$$

Combining with assumption 3 in Theorem 1, we have $1 - 2\lambda\eta - \sqrt{\frac{2L}{l}} \cdot 2\lambda\eta > 0$, which means

$$0 < 1 - 2\lambda\eta - \frac{L}{x^* x_t}. \tag{46}$$

Combining with Eq.(44), (43), (46), if $t > T(\alpha, \lambda, \eta, l, x_0)$, we have

$$\mathbb{E}[(x_{t+1} - x^*)^2 | x_t] < (1 - 2\lambda\eta)^2 (x_t - x^*)^2 + \frac{4V\eta^3\lambda}{l}. \tag{47}$$

Considering the expectation with respect to the distribution of $x_t$, we have

$$\mathbb{E}(x_{t+1} - x^*)^2 < (1 - 2\lambda\eta)^2 \mathbb{E}(x_t - x^*)^2 + \frac{4V\eta^3\lambda}{l}. \tag{48}$$

Approximate $(1 - 2\lambda\eta)^2 = 1 - 4\lambda\eta$, and iterate Eq.(48) for $t - T(\alpha, \lambda, \eta, l, x_0)$ times, we have

$$\mathbb{E}(x_t - x^*)^2 < (1 - 4\lambda\eta)^{t - T(\alpha,\lambda,\eta,l,x_0)} \mathbb{E}(x_{T(\alpha,\lambda,\eta,l,x_0)} - x^*)^2 + \frac{V\eta^2}{l}. \tag{49}$$

$T(\alpha, \lambda, \eta, l, x_0)$ is finite, and only depends on $\alpha, \lambda, \eta, l, x_0$. Hence we can set $B = max\{(1 - 4\lambda\eta)^{-t}\mathbb{E}(x_t - x^*)^2 | t = 0, 1, 2, ..., T(\alpha, \lambda, \eta, l, x_0)\}$, note $B$ is finite by

$$\mathbb{E}x_{t+1}^2 = \mathbb{E}[(1-2\lambda\eta)^2 x_t^2 + 2(1-2\lambda\eta)L_t + \frac{L_t^2}{x_t^2}] < (1-4\lambda\eta)\mathbb{E}x_t^2 + 2(1-2\lambda\eta)L + \frac{V + L^2}{min(x_0^2, l\eta/(2\lambda))}. \tag{50}$$

therefore, $\forall t > 0$, we have

$$\mathbb{E}(x_t - x^*)^2 < (1 - 4\lambda\eta)^t B + \frac{V\eta^2}{l}. \tag{51}$$

$\square$

## B.3 PROOF OF THEOREM 2

**Lemma 3.** *Assume* $\alpha, \beta, \varepsilon \in (0, 1)$, *where* $\beta << 1$. *Denote* $diag(1 - \frac{2\beta}{1-\alpha}, \alpha, \alpha^2 + \frac{2\alpha^2}{1-\alpha}\beta$ *as* $\mathbf{\Lambda}$, $\mathbf{k} = (\frac{1}{(1-\alpha)^2}, -\frac{2\alpha}{(1-\alpha)^2}, \frac{\alpha^2}{(1-\alpha)^2})^T$, $\mathbf{e} = (1, 1, 1)^T$. *If* $\varepsilon < \frac{1}{3}[\frac{1-\alpha^2}{\beta} - \frac{8}{1-\alpha}]$, *then* $\forall \mathbf{d} \in \mathbb{R}^p$, *we have*

$$||(\mathbf{\Lambda} - \varepsilon\beta(1-\alpha)^2 \mathbf{k}\mathbf{e}^T)\mathbf{d}||_2^2 < (1 - \frac{4\beta}{1-\alpha})||\mathbf{d}||_2^2 \tag{52}$$

*Proof.* Omit $O(\beta^2)$ part, we have

$$||(\mathbf{\Lambda} - \varepsilon\beta(1-\alpha)^2\mathbf{k}\mathbf{e}^T)\mathbf{d}||_2^2 = (1 - \frac{4\beta}{1-\alpha})d_1^2 + \alpha^2 d_2^2 + \alpha^4(1 + \frac{4\beta}{1-\alpha})d_3^2 - 2\varepsilon\beta(d_1 + d_2 + d_3)(d_1 - 2\alpha^2 d_2 + \alpha^4 d_3) \tag{53}$$

$$
\begin{aligned}
&(d_1 + d_2 + d_3)(d_1 - 2\alpha^2 d_2 + \alpha^4 d_3) \\
&= \frac{[d_1 + (1 - 2\alpha^2)d_2]^2}{2} + \frac{[d_1 + (1 + \alpha^4)d_3]^2}{2} - (1/2 + 2\alpha^4)d_2^2 - \frac{1 + \alpha^8}{2} + (\alpha^4 - 2\alpha^2)d_2 d_3 \\
&\geq -(\frac{1}{2} + 2\alpha^4 + \frac{\alpha^4}{2})d_2^2 - (\frac{\alpha^8}{2} + \frac{\alpha^4}{2} - 2\alpha^2 + \frac{5}{2})d_3^2 \\
&\geq -3d_2^2 - \frac{5}{2}d_3^2.
\end{aligned}
$$

$$\tag{54}$$

Then we have

$$||(\mathbf{\Lambda} - \varepsilon\beta(1-\alpha)^2 \mathbf{k}\mathbf{e}^T)\mathbf{d}||_2^2 \leq (1 - \frac{4\beta}{1-\alpha})d_1^2 + (\alpha^2 + 3\beta\varepsilon)d_2^2 + (\alpha^4 + \frac{4\beta\alpha^4}{1-\alpha} + \frac{5\beta\varepsilon}{2})d_3^2. \quad (55)$$

Since $\varepsilon < \frac{1}{3}[\frac{1-\alpha^2}{\beta} - \frac{8}{1-\alpha}]$, we have

$$\alpha^2 + 3\beta\varepsilon \quad < \quad 1 - \frac{4\beta}{1-\alpha}, \quad (56)$$

$$\alpha^4 + \frac{4\beta\alpha^4}{1-\alpha} + \frac{5\beta\varepsilon}{2} \quad < \quad 1 - \frac{4\beta}{1-\alpha}. \quad (57)$$

Hence, we have

$$||(\mathbf{\Lambda} - \varepsilon\beta(1-\alpha)^2 \mathbf{k}\mathbf{e}^T)\mathbf{d}||_2^2 < (1 - \frac{4\beta}{1-\alpha})||\mathbf{d}||_2^2 \quad (58)$$

$$\square$$

*Proof of Theorem 2.* The update rule is

$$\begin{aligned}
\mathbf{w}_{t+1} &= \mathbf{w}_t - \eta\mathbf{v}_t \\
&= \mathbf{w}_t - \eta(\alpha\mathbf{v}_{t-1} + \frac{\tilde{\mathbf{g}}_t}{||\mathbf{w}_t||} + \lambda\mathbf{w}_t) \\
&= \mathbf{w}_t - \eta(\alpha\frac{\mathbf{w}_{t-1} - \mathbf{w}_t}{\eta} + \frac{\tilde{\mathbf{g}}_t}{||\mathbf{w}_t||} + \lambda\mathbf{w}_t) \\
&= (1 - \eta\lambda + \alpha)\mathbf{w}_t - \alpha\mathbf{w}_{t-1} - \mathbf{g}_t\eta.
\end{aligned} \quad (59)$$

Then we have

$$\begin{aligned}
||\mathbf{w}_{t+1}||_2^2 &= (1 - \eta\lambda + \alpha)^2||\mathbf{w}_t||_2^2 - 2\alpha(1 + \alpha - \eta\lambda)\langle\mathbf{w}_t, \mathbf{w}_{t-1}\rangle + \alpha^2||\mathbf{w}_{t-1}||_2^2 \\
&\quad + ||\mathbf{g}_t||_2^2\eta^2 + 2\langle\alpha\mathbf{w}_{t-1}, \mathbf{g}_t\eta\rangle \\
&= (1 - \eta\lambda + \alpha)^2||\mathbf{w}_t||_2^2 - 2\alpha(1 + \alpha - \eta\lambda)\langle\mathbf{w}_t, \mathbf{w}_{t-1}\rangle + \alpha^2||\mathbf{w}_{t-1}||_2^2 \\
&\quad + ||\mathbf{g}_t||_2^2\eta^2 + 2\langle\alpha(\mathbf{w}_t + \eta\mathbf{v}_{t-1}), \mathbf{g}_t\eta\rangle \\
&= (1 - \eta\lambda + \alpha)^2||\mathbf{w}_t||_2^2 - 2\alpha(1 + \alpha - \eta\lambda)\langle\mathbf{w}_t, \mathbf{w}_{t-1}\rangle + \alpha^2||\mathbf{w}_{t-1}||_2^2 \\
&\quad + \frac{\tilde{h}_t\eta^2}{||\mathbf{w}_t||_2^2}.
\end{aligned} \quad (60)$$

$$\langle\mathbf{w}_{t+1}, \mathbf{w}_t\rangle = (1 + \alpha - \lambda\eta)||\mathbf{w}_t||^2 - \alpha\langle\mathbf{w}_t, \mathbf{w}_{t-1}\rangle, \quad (61)$$

Let $\mathbf{X}_t, \mathbf{A}, \mathbf{e}$ denote:

$$\mathbf{X}_t = \begin{pmatrix} a_t \\ b_t \\ c_t \end{pmatrix} = \begin{pmatrix} ||\mathbf{w}_t||_2^2 \\ \langle\mathbf{w}_t, \mathbf{w}_{t-1}\rangle \\ ||\mathbf{w}_{t-1}||_2^2 \end{pmatrix}, \quad (62)$$

$$\mathbf{A} = \begin{pmatrix} (1 + \alpha - \lambda\eta)^2 & -2\alpha(1 + \alpha - \lambda\eta) & \alpha^2 \\ 1 + \alpha - \lambda\eta & -\alpha & 0 \\ 1 & 0 & 0 \end{pmatrix}, \quad (63)$$

$$\mathbf{e} = \begin{pmatrix} 1 \\ 0 \\ 0 \end{pmatrix}, \quad (64)$$

$$(65)$$

respectively. The Eq.(60), (61) is formulated as a iterative map:

$$\mathbf{X}_{t+1} = \mathbf{A}\mathbf{X}_t + \frac{\tilde{h}_t\eta^2}{\mathbf{e}^T\mathbf{X}_t}\mathbf{e}. \quad (66)$$

When $\lambda\eta < (1 - \sqrt{\alpha})^2$, the eigen value of $\boldsymbol{A}$ are all real number:

$$\lambda_1 = \frac{(1+\alpha-\lambda\eta)^2 + (1+\alpha-\lambda\eta)\sqrt{(1+\alpha-\lambda\eta)^2 - 4\alpha}}{2} - \alpha = 1 - \frac{2\lambda\eta}{1-\alpha} + O(\lambda^2\eta^2), \quad (67)$$

$$\lambda_2 = \alpha, \quad (68)$$

$$\lambda_3 = \frac{(1+\alpha-\lambda\eta)^2 - (1+\alpha-\lambda\eta)\sqrt{(1+\alpha-\lambda\eta)^2 - 4\alpha}}{2} - \alpha = \alpha^2 + \frac{2\alpha^2}{(1-\alpha)}\lambda\eta + O(\lambda^2\eta^2) \quad (69)$$

and they satisfy

$$0 < \lambda_3 < \lambda_2 = \alpha < \lambda_1 < 1. \quad (70)$$

Therefore, $\boldsymbol{A}$ can be formulated as

$$\boldsymbol{S}^{-1}\boldsymbol{A}\boldsymbol{S} = \boldsymbol{\Lambda}, \quad (71)$$

where $\boldsymbol{\Lambda}$ is a diagonal matrix whose diagonal elements are the eigen value of $\boldsymbol{A}$; the column vector of $\boldsymbol{S}$ is the eigen vectors of $\boldsymbol{A}$, note the fromulation of $\boldsymbol{S}, \boldsymbol{\Lambda}$ are not unique. Specifically, we set $\boldsymbol{\Lambda}, \boldsymbol{S}$ as

$$\boldsymbol{\Lambda} = \begin{pmatrix} \lambda_1 & 0 & 0 \\ 0 & \lambda_2 & 0 \\ 0 & 0 & \lambda_3 \end{pmatrix}, \quad (72)$$

$$\boldsymbol{S} = \begin{pmatrix} 1 & 1 & 1 \\ \frac{1+\alpha-\lambda\eta}{\alpha+\lambda_1} & \frac{1+\alpha-\lambda\eta}{\alpha+\lambda_2} & \frac{1+\alpha-\lambda\eta}{\alpha+\lambda_3} \\ \frac{1}{\lambda_1} & \frac{1}{\lambda_2} & \frac{1}{\lambda_3} \end{pmatrix}. \quad (73)$$

Moreover, the inverse of $\boldsymbol{S}$ exists, and can be explicitly expressed as

$$\boldsymbol{S}^{-1} = \begin{pmatrix} \frac{(\alpha+\lambda_1)\lambda_1}{(\lambda_1-\alpha)(\lambda_1-\lambda_3)} & -\frac{2\lambda_1(\alpha+\lambda_1)(\alpha+\lambda_3)}{(\lambda_1-\lambda_3)(\lambda_1-\alpha)(1+\alpha-\beta)} & \frac{\lambda_1\lambda_3(\alpha+\lambda_1)}{(\lambda_1-\alpha)(\lambda_1-\lambda_3)} \\ -\frac{2\alpha^2}{(\lambda_1-\alpha)(\alpha-\lambda_3)} & \frac{2\alpha(\alpha+\lambda_1)(\alpha+\lambda_3)}{(\lambda_1-\alpha)(\alpha-\lambda_3)(1+\alpha-\beta)} & -\frac{2\alpha\lambda_1\lambda_3}{(\lambda_1-\alpha)(\alpha-\lambda_3)} \\ \frac{(\alpha+\lambda_3)\lambda_3}{(\alpha-\lambda_3)(\lambda_1-\lambda_3)} & -\frac{2\lambda_3(\alpha+\lambda_3)(\alpha+\lambda_1)}{(\lambda_1-\lambda_3)(\alpha-\lambda_3)(1+\alpha-\beta)} & \frac{(\alpha+\lambda_3)\lambda_1\lambda_3}{(\alpha-\lambda_3)(\lambda_1-\lambda_3)} \end{pmatrix}. \quad (74)$$

let $\boldsymbol{Y}_t = \boldsymbol{S}^{-1}\boldsymbol{X}_t$, combining with Eq.(66), we have

$$\boldsymbol{Y}_{t+1} = \boldsymbol{\Lambda}\boldsymbol{Y}_t + \frac{\tilde{h}_t\eta^2}{(\boldsymbol{S}^T\boldsymbol{e})^T\boldsymbol{Y}_t}\boldsymbol{S}^{-1}\boldsymbol{e}. \quad (75)$$

Combining with Eq.(73) and Eq.(74), and set $\boldsymbol{Y}_t = (\tilde{a}_t, \tilde{b}_t, \tilde{c}_t)^T$, we rewrite Eq.(75) as

$$\tilde{a}_{t+1} = \lambda_1\tilde{a}_t + \frac{\tilde{h}_t\eta^2}{\tilde{a}_t + \tilde{b}_t + \tilde{c}_t} \cdot \frac{(\alpha+\lambda_1)\lambda_1}{(\lambda_1-\alpha)(\lambda_1-\lambda_3)}, \quad (76)$$

$$\tilde{b}_{t+1} = \alpha\tilde{b}_t - \frac{\tilde{h}_t\eta^2}{\tilde{a}_t + \tilde{b}_t + \tilde{c}_t} \cdot \frac{2\alpha^2}{(\lambda_1-\alpha)(\alpha-\lambda_3)}, \quad (77)$$

$$\tilde{c}_{t+1} = \lambda_3\tilde{c}_t + \frac{\tilde{h}_t\eta^2}{\tilde{a}_t + \tilde{b}_t + \tilde{c}_t} \cdot \frac{(\alpha+\lambda_3)\lambda_3}{(\alpha-\lambda_3)(\lambda_1-\lambda_3)}. \quad (78)$$

Note $||\boldsymbol{w}_t||_2^2 = \tilde{a}_t + \tilde{b}_t + \tilde{c}_t$. Now we prove the following inequations by mathematical induction

$$\tilde{b}_t < 0, \quad (79)$$

$$\tilde{c}_t > 0, \quad (80)$$

$$(\alpha - \lambda_1)\tilde{b}_t > (\lambda_1 - \lambda_3)\tilde{c}_t, \quad (81)$$

$$\tilde{a}_t + \tilde{b}_t + \tilde{c}_t > 0, \quad (82)$$

$$\tilde{a}_{t+1} + \tilde{b}_{t+1} + \tilde{c}_{t+1}, > \lambda_1(\tilde{a}_t + \tilde{b}_t + \tilde{c}_t) + \frac{\tilde{h}_t\eta^2}{\tilde{a}_t + \tilde{b}_t + \tilde{c}_t}, \quad (83)$$

Since the start point $X_1 = (a_1, a_1, a_1)^T$ $(a_1 > 0)$, the start point $Y_1 = S^{-1}X_1$. Combining with Eq.(74), we have

$$\tilde{b}_1 = -\frac{2\alpha^2\lambda\eta}{(\lambda_1 - \alpha)(\alpha - \lambda_3)}a_1, \tag{84}$$

$$\tilde{c}_1 = \frac{\lambda_3(\lambda_3 + \alpha)(1 - \alpha + \lambda\eta)}{(\lambda_3 - \alpha)(\lambda_1 - \lambda_3)(1 + \alpha - \lambda\eta)}\left(\frac{1 - \alpha - \lambda\eta}{1 - \alpha + \lambda\eta} - \lambda_1\right)a_1, \tag{85}$$

by which we have $(\alpha - \lambda_1)\tilde{b}_1 > (\lambda_1 - \lambda_3)\tilde{c}_1$. Besides $\tilde{a}_1 + \tilde{b}_1 + \tilde{c}_1 = e^T S Y_1 = e^T X_1 = a_1 > 0$.

Suppose for $t = T$, Eq. (79), (80), (81), (82) hold, combining with Eq.(77), (78), we can derive $\tilde{b}_{T+1} < 0, \tilde{a}_{T+1} > 0$, so Eq.(79), (80) hold for $t = T + 1$; Besides, we have

$$(\alpha - \lambda_1)\tilde{b}_{T+1} = \alpha(\alpha - \lambda_1)\tilde{b}_T + \frac{\tilde{h}_t\eta^2}{\tilde{a}_T + \tilde{b}_T + \tilde{c}_T} \cdot \frac{2\alpha^2}{(\alpha - \lambda_3)}$$
$$> \lambda_3(\lambda_1 - \lambda_3)\tilde{c}_T + \frac{\tilde{h}_t\eta^2}{\tilde{a}_T + \tilde{b}_T + \tilde{c}_T} \cdot \frac{(\alpha + \lambda_3)\lambda_3}{(\alpha - \lambda_3)} \tag{86}$$
$$= (\lambda_1 - \lambda_3)\tilde{c}_{T+1},$$

thus Eq.(81) holds for $t = T + 1$. Sum Eq.(76), Eq.(77), Eq.(78), due to Eq.(81) we have

$$\tilde{a}_{T+1} + \tilde{b}_{T+1} + \tilde{c}_{T+1} = \lambda_1\tilde{a}_T + \alpha\tilde{b}_T + \lambda_3\tilde{c}_T + \frac{\tilde{h}_t\eta^2}{\tilde{a}_T + \tilde{b}_T + \tilde{c}_T}$$
$$> \lambda_1(\tilde{a}_T + \tilde{b}_T + \tilde{c}_T) + \frac{\tilde{h}_t\eta^2}{\tilde{a}_T + \tilde{b}_T + \tilde{c}_T}, \tag{87}$$

Eq.(83) holds for $t = T + 1$, combining with the fact that $\tilde{a}_T + \tilde{b}_T + \tilde{c}_T > 0$, we have $\tilde{a}_{T+1} + \tilde{b}_{T+1} + \tilde{c}_{T+1} > 0$.

According to Lemma 2, we can estimate the lower bound of $\tilde{a}_t + \tilde{b}_t + \tilde{c}_t$: when $t > 1 + \frac{log((\sqrt{2}-1)\sqrt{l\eta/(4\lambda)}) - log(|||w_0||_2^2 - \sqrt{l\eta/(4\lambda)}|)}{log(1 - 2\lambda\eta)}$,

$$\tilde{a}_t + \tilde{b}_t + \tilde{c}_t \geq \sqrt{\frac{l\eta}{2(1 - \lambda_1)}} \approx \sqrt{\frac{l\eta(1 - \alpha)}{4\lambda}} \tag{88}$$

Now we can analyze the expectation of distance($l^2$ norm) between $Y_t = (\tilde{a}_t, \tilde{b}_t, \tilde{c}_t)^T$ and the fixed point $Y^* = (\tilde{a}^*, \tilde{b}^*, \tilde{c}^*)^T$ which satisfies

$$Y^* = \Lambda Y^* + \frac{L\eta^2}{(S^T e)^T Y^*}S^{-1}e. \tag{89}$$

Assume $x_t = \tilde{a}_t + \tilde{b}_t + \tilde{c}_t$, $x^* = \tilde{a}^* + \tilde{b}^* + \tilde{c}^* > 0$, then we have:

$$Y_{t+1} - Y^* = (\Lambda - \frac{L\eta^2}{x_t x^*}ke^T)(Y_t - Y^*) + \frac{(\tilde{h}_t - L)\eta^2}{x_t}k \tag{90}$$

where $k = (k_1, k_2, k_3)^T = S^{-1}e$. In the following context, we will omit the $O(\lambda^2\eta^2)$ part since $\lambda\eta \ll 1$. $k_1, k_2, k_3$ can be approximated as

$$k_1 = \frac{1}{(1 - \alpha)^2} + O(\beta), \tag{91}$$

$$k_2 = -\frac{2\alpha}{(1 - \alpha)^2} + O(\beta), \tag{92}$$

$$k_3 = \frac{\alpha^2}{(1 - \alpha)^2} + O(\beta). \tag{93}$$

Then we have

$$\mathbb{E}[||Y_{t+1} - Y^*||_2^2|Y_t] = ||(\Lambda - \frac{L}{x_t x^*}ke^T)(Y_t - Y^*)||_2^2 + \frac{\mathbb{E}[(\tilde{h}_t - L)^2|Y_t]\eta^4}{x_t^2}||k||_2^2 \tag{94}$$

The fixed point of Eq.(89) is computed as $x^* = \tilde{a}^* + \tilde{b}^* + \tilde{c}^* = \sqrt{\frac{L\eta}{\lambda(1-\alpha)(2-\frac{\lambda\eta}{1+\alpha})}}$, and we have known that if $t > 1 + \frac{log((\sqrt{2}-1)\sqrt{l\eta/(4\lambda)})-log(|||\boldsymbol{w}_0||_2^2-\sqrt{l\eta/(4\lambda)}|)}{log(1-2\lambda\eta)}$, $x_t > \sqrt{\frac{l\eta(1-\alpha)}{4\lambda}}$,, therefore we have

$$\frac{L\eta^2}{x_t x^*} \quad < \quad \sqrt{\frac{2L}{l}} \cdot 2\lambda\eta, \tag{95}$$

$$\frac{\mathbb{E}[(\tilde{h}_t - L)^2|\boldsymbol{Y_t}]}{x_t^2}\eta^4 \quad < \quad \frac{4V\eta^3\lambda}{l(1-\alpha)}. \tag{96}$$

According to assumption 3, we can prove

$$\sqrt{\frac{2L}{l}} < \frac{(1-\alpha)^2}{3}[\frac{1-\alpha^2}{\beta} - \frac{8}{1-\alpha}], \tag{97}$$

combining with Lemma 3, we have

$$\mathbb{E}[||\boldsymbol{Y}_{t+1} - \boldsymbol{Y}^*||_2^2|\boldsymbol{Y_t}] < (1 - \frac{4\lambda\eta}{1-\alpha})||\boldsymbol{Y}_t - \boldsymbol{Y}^*||_2^2 + \frac{4V\eta^3\lambda||\boldsymbol{k}||_2^2}{l(1-\alpha)}, \tag{98}$$

which implies

$$\mathbb{E}||\boldsymbol{Y}_{t+1} - \boldsymbol{Y}^*||_2^2 < (1 - \frac{4\lambda\eta}{1-\alpha})^{t-T}\mathbb{E}||\boldsymbol{Y}_T - \boldsymbol{Y}^*||_2^2 + \frac{V\eta^2(1+4\alpha^2+\alpha^4)}{l(1-\alpha)^4}, \tag{99}$$

where $T = [1 + \frac{log((\sqrt{2}-1)\sqrt{l\eta/(4\lambda)})-log(|||\boldsymbol{w}_0||_2^2-\sqrt{l\eta/(4\lambda)}|)}{log(1-2\lambda\eta)}]$. Therefore, similar to the proof of theorem 1, $\exists B > 0$,

$$\mathbb{E}||\boldsymbol{Y}_{t+1} - \boldsymbol{Y}^*||_2^2 < (1 - \frac{4\lambda\eta}{1-\alpha})^t B + \frac{V\eta^2(1+4\alpha^2+\alpha^4)}{l(1-\alpha)^4}. \tag{100}$$

Recall $||\boldsymbol{w}_t||_2^2 = \boldsymbol{e}^T\boldsymbol{Y}_t$, therefore

$$\mathbb{E}[||\boldsymbol{w}_t||_2^2 - (w^*)^2]^2 \le 3\mathbb{E}||\boldsymbol{Y}_{t+1} - \boldsymbol{Y}^*||_2^2 < 3(1 - \frac{4\lambda\eta}{1-\alpha})^t B + \frac{3V\eta^2(1+4\alpha^2+\alpha^4)}{l(1-\alpha)^4}. \tag{101}$$

$\square$

**Remark 5.** *By Eq.(101), the variance of $||\boldsymbol{w}_t||_2^2$ is bounded by $\frac{3V\eta^2(1+4\alpha^2+\alpha^4)}{l(1-\alpha)^4}$, which is not small enough. But we somehow can reduce it: according to Eq.(96), if $x_t$ is close to its theoretical value, then $\frac{\mathbb{E}[(\tilde{h}_t-L)^2|\boldsymbol{Y}_t]}{x_t^2}\eta^4 < \frac{4V\eta^3\lambda(1-\alpha)}{l}$, hence variance of $||\boldsymbol{w}_t||_2^2$ can be bounded by $\frac{3V\eta^2(1+4\alpha^2+\alpha^4)}{2L(1-\alpha)^2}$.*

### B.4 Proof of Theorem 3

*Proof.* In SGD case, we have

$$\langle\boldsymbol{w}_{t+1}, \boldsymbol{w}_t\rangle = (1 - \lambda\eta)||\boldsymbol{w}_t||_2^2, \tag{102}$$

then we have

$$\cos^2 \Delta_t = \frac{\langle\boldsymbol{w}_{t+1}, \boldsymbol{w}_t\rangle^2}{||\boldsymbol{w}_t||_2^2 \cdot ||\boldsymbol{w}_{t+1}||_2^2} = (1 - 2\lambda\eta)\frac{||\boldsymbol{w}_t||_2^2}{||\boldsymbol{w}_{t+1}||_2^2}. \tag{103}$$

According to the definition of $\Delta_t$, $\Delta_t \ge 0$, and $\Delta_t$ is very close to 0, hence we have

$$\Delta_t = \sin \Delta_t \quad = \quad \sqrt{1 - \cos^2 \Delta_t} \tag{104}$$

$$= \quad \sqrt{1 - (1-\lambda\eta)^2\frac{||\boldsymbol{w}_t||_2^2}{||\boldsymbol{w}_{t+1}||_2^2}} \tag{105}$$

$$= \quad \sqrt{1 - (1-\lambda\eta)^2\frac{x_t}{x_{t+1}}} \tag{106}$$

where $x_t, x_{t+1}$ denotes $||\boldsymbol{w}_t||_2^2, ||\boldsymbol{w}_{t+1}||_2^2$ respectively as in Eq. (42). Assume $t$ is sufficiently large so that $x_t, x_{t+1}$ are close to $x^* = \sqrt{\frac{L\eta}{2\lambda}}$, the first order of Taylor series expansion of Eq.(106) at $x_t = x_{t+1} = x^*$ is

$$\Delta_t = \sqrt{2\lambda\eta} + \frac{(1-\lambda\eta)^2}{2\sqrt{2\lambda\eta}} \cdot \frac{1}{x^*} \cdot [(x_{t+1} - x^*) - (x_t - x^*)]. \tag{107}$$

Reorganizing Eq.(107), and applying Cauchy Inequality, we have

$$|\Delta_t - \sqrt{2\lambda\eta}|^2 \quad = \quad \frac{(1-\lambda\eta)^4}{8\lambda\eta} \cdot \frac{1}{(x^*)^2} \cdot [(x_{t+1} - x^*) - (x_t - x^*)]^2 \tag{108}$$

$$\leq \quad \frac{1}{8\lambda\eta} \cdot \frac{2\lambda}{L\eta} \cdot 2[(x_t - x^*)^2 + (x_{t+1} - x^*)^2] \tag{109}$$

$$= \quad \frac{1}{2L\eta^2}[(x_t - x^*)^2 + (x_{t+1} - x^*)^2] \tag{110}$$

Combining with Eq.(51) and Eq.(108), we have

$$\mathbb{E}|\Delta_t - \sqrt{2\lambda\eta}|^2 = \frac{1}{2L\eta^2}[\mathbb{E}(x_t - x^*)^2 + \mathbb{E}(x_{t+1} - x^*)^2] \leq (1-4\lambda\eta)^t \cdot C + \frac{V}{Ll}, \tag{111}$$

where $C = \frac{1}{2L\eta^2} \cdot B$, $B$ is defined in Eq.(51).

In SGDM case, the angular update can be computed by

$$\Delta_t \quad = \quad \sin(\Delta_t) \tag{112}$$

$$= \quad \sqrt{1 - \cos^2 \Delta_t} \tag{113}$$

$$= \quad \sqrt{1 - \frac{\langle \boldsymbol{w}_t, \boldsymbol{w}_{t+1} \rangle^2}{||\boldsymbol{w}_t||_2^2 \cdot ||\boldsymbol{w}_{t+1}||_2^2}} \tag{114}$$

$$= \quad \sqrt{1 - \frac{b_t^2}{a_t c_t}}, \tag{115}$$

where $(a_t, b_t, c_t) = (||\boldsymbol{w}_t||_2^2, \langle \boldsymbol{w}_t, \boldsymbol{w}_{t+1} \rangle, ||\boldsymbol{w}_{t+1}||_2^2)$. According to the proof of theorem 2, when t is sufficiently large, $(a_t, b_t, c_t)$ will be close to the fixed point of Eq.(66), $(a^*, b^*, c^*)$, where $a^* = c^* = (\sqrt{\frac{L\eta}{\lambda(1-\alpha)(2-\frac{\lambda\eta}{1+\alpha})}}, b^* = \frac{1+\alpha-\lambda}{1+\alpha}a^*$. Then the first order Taylor series expansion of Eq.(115) at $(a_t = a^*, b_t = b^*, c_t = c^*)$ is

$$\Delta_t \quad = \quad \sqrt{1 - \frac{b_t^2}{a_t c_t}} \tag{116}$$

$$= \quad \sqrt{\frac{2\lambda\eta}{1+\alpha}} + \frac{\sqrt{1+\alpha}}{2\sqrt{2\lambda\eta}} \cdot (1 - \frac{\lambda\eta}{1+\alpha})^2 [\frac{a_t - a^*}{a^*} - \frac{2(b_t - b^*)}{b^*} + \frac{c_t - c^*}{c^*}]. \tag{117}$$

Now substituting $(a_t, b_t, c_t)$ with $(\tilde{a}_t, \tilde{b}_t, \tilde{c}_t)$ defined in Eq.(75, 76, 77, 78), Eq.(107) is rewritten as

$$\Delta_t = \sqrt{\frac{2\lambda\eta}{1+\alpha}} + \frac{\sqrt{1+\alpha}}{2\sqrt{2\lambda\eta}} \cdot (1 - \frac{\lambda\eta}{1+\alpha})^2 \frac{1}{a^*}[\frac{(1-\alpha)^2}{\alpha^2}(\tilde{c}_t - \tilde{c}^*) + O(\lambda\eta)] \tag{118}$$

where $(\tilde{a}^*, \tilde{b}^*, \tilde{c}^*)$ is the fixed point of Eq.(89). Omit $O(\lambda\eta)$, we have

$$|\Delta_t - \sqrt{\frac{2\lambda\eta}{1+\alpha}}|^2 \quad = \quad \frac{1+\alpha}{8\lambda\eta} \cdot (1 - \frac{\lambda\eta}{1+\alpha})^4 \frac{\lambda(1-\alpha)(2-\frac{\lambda\eta}{1+\alpha})}{L\eta} [\frac{(1-\alpha)^4}{\alpha^4}(\tilde{c} - \tilde{c}^*)^2] \tag{119}$$

$$\leq \quad \frac{(1-\alpha^2)(1-\alpha)^4}{4L\eta^2\alpha^4}||\boldsymbol{Y}_t - \boldsymbol{Y}^*||_2^2, \tag{120}$$

where $\boldsymbol{Y}_t$, $\boldsymbol{Y}^*$ is defined in Eq.(75, 89) respectively. According to Eq.(100), the mean square error $\mathbb{E}|\Delta_t - \sqrt{\frac{2\lambda\eta}{1+\alpha}}|^2$ can be bounded by

$$\mathbb{E}|\Delta_t - \sqrt{\frac{2\lambda\eta}{1+\alpha}}|^2 \leq \frac{(1-\alpha^2)(1-\alpha)^4}{4L\eta^2\alpha^4}\mathbb{E}||\boldsymbol{Y}_t - \boldsymbol{Y}^*||_2^2 \tag{121}$$

$$\leq (1 - \frac{4\lambda\eta}{1-\alpha})^t C + \frac{(1-\alpha^2)(1-\alpha)^4}{4L\eta^2\alpha^4} \cdot \frac{V\eta^2(1+4\alpha^2+\alpha^4)}{l(1-\alpha)^4} \tag{122}$$

$$= (1 - \frac{4\lambda\eta}{1-\alpha})^t C + \frac{V(1-\alpha^2)(1+4\alpha^2+\alpha^4)}{4Ll\alpha^4}, \tag{123}$$

where $C = \frac{(1-\alpha^2)(1-\alpha)^4}{4L\eta^2\alpha^4} \cdot B$, $B$ is defined in Eq.(101). $\qquad\square$

## B.5 Proof of Corollary 3.1

*Proof of Corollary 3.1.* In SGD case, and Eq.(42), we have

$$||\boldsymbol{w}_{t+1}||_2^2 > (1 - 2\lambda\eta)||\boldsymbol{w}_t||_2^2, \tag{124}$$

which means

$$||\boldsymbol{w}_{t+T}||_2^2 > (1 - 2\lambda\eta)^T ||\boldsymbol{w}_t||_2^2. \tag{125}$$

On the other hand, we know that when $\eta$ is divided by $k$, $||\boldsymbol{w}_t||_2^2$ should be divided by $\sqrt{k}$ to reach the new equilibrium condition, therefore we have

$$\frac{||\boldsymbol{w}_{t+T}||^2}{||\boldsymbol{w}_t||^2} = \frac{1}{\sqrt{k}} > (1 - 2\lambda\eta)^T. \tag{126}$$

Since $\lambda\eta \ll 1$, $\log(1 - 2\lambda\eta) \approx -2\lambda\eta$, thus

$$T > \frac{\log(k)}{4\lambda\eta}. \tag{127}$$

.

In SGDM case, by Eq.(87), we have

$$||\boldsymbol{w}_{t+1}||_2^2 > (1 - \frac{2\lambda\eta}{1-\alpha})||\boldsymbol{w}_t||_2^2, \tag{128}$$

Similar to SGD case, we have

$$T > \frac{\log(k)(1-\alpha)}{4\lambda\eta}. \tag{129}$$

. $\qquad\square$

## C EXPERIMENTS

### C.1 EXPERIMENTS ON SYTHETIC DATA

In this section we apply experiments on sythetic data to prove our claim in section 4: theorem 1 only requires expected square norm of unit gradient is locally steady, the expectations do not need to remain unchanged across the whole training process. The proof of theorem implies square norm of weight is determined by the following iterative map:

$$x_{t+1} = (1 - 2\lambda\eta)x_t + \frac{L_t\eta^2}{x_t}, \tag{130}$$

where $\lambda, \eta \in (0,1)$, $L_t$ denotes the square of unit gradient norm. Hence we simulate $x_t$ with different type of $\{L_t\}_{t=1}^{\infty}$. Results in Figure 4 shows as long as the local variance of square norm of unit gradient is not too much, and expectation of $L_t$ changes smoothly, weight norm can quickly converge to its theoretical value base on expectation of square norm of unit gradient.

We also simulate SGDM case by following iteration map

$$\boldsymbol{X}_{t+1} = \boldsymbol{A}\boldsymbol{X}_t + \frac{L_t\eta^2}{\boldsymbol{X}_t[0]} \cdot \boldsymbol{e}, \tag{131}$$

where $\boldsymbol{A}$, $\boldsymbol{X}_t$, $\boldsymbol{e}$ is defined as Eq.(62), (63), (64). Simulation results is shown in Figure 5.

### C.2 COMPLEMENTARY IN MULTI-STAGE LEARNING RATE SCHEDULE

In this section we present complementary results in Multi-Stage Learning Rate Schedule experiment.

The plots of weight norm (empirical and predicted values) in multi-learning rate stage is shown in Figure 6. We also present the test performance of resent50/maskrcnn on Imagenet/MSCOCO with multi-stage learning rate schedule mentioned in Section 5.2. We only provide complementary results for reference only. **We do not intend to prove the advantages or disadvantages of rescaling strategy here, it is beyond the discussion of this paper**.

|          | Top 1 Accuracy(%) |
|----------|-------------------|
| Standard | 76.25             |
| Rescale  | 76.27             |

Table 1: Performance of Resnet50 on Imagenet with multi-stage learning rate scheduler.

|          | $AP^{bbox}$ | $AP^{bbox}_{50}$ | $AP^{bbox}_{75}$ | $AP^{mask}$ | $AP^{mask}_{50}$ | $AP^{mask}_7$ |
|----------|-------------|------------------|------------------|-------------|------------------|---------------|
| Standard | 39.25       | 58.88            | 42.95            | 35.16       | 56.09            | 37.48         |
| Rescale  | 38.38       | 57.98            | 42.31            | 34.49       | 55.27            | 36.71         |

Table 2: Performance of Resnet50 on Imagenet with multi-stage learning rate scheduler in Section.

### C.3 RETHINKING LINEAR SCALING PRINCIPLE IN SPHERICAL MOTION DYNAMICS

In this section, we will discuss the effect of Linear Scaling Principle (LSP) under the view of SMD. Linear Scaling Principle is proposed by Goyal et al. (2017) to tune the learning rate $\eta$ with batch size $B$ by $\eta \propto B$. The intuition of LSP is if weights do not change too much within $k$ iterations, then $k$ iterations of SGD with learning rate $\eta$ and minibatch size $B$ (Eq.(132)) can be approximated by a single iteration of SGD with learning rate $k\eta$ and minibatch size $kB$ (Eq.(133):

$$\boldsymbol{w}_{t+k} = \boldsymbol{w}_t - \eta\sum_{j<k}\Big(\frac{1}{B}\sum_{x\in\mathcal{B}_j}\frac{\partial \boldsymbol{L}}{\partial \boldsymbol{w}}\Big|_{\boldsymbol{w}_{t+j},x} + \lambda\boldsymbol{w}_{t+j}\Big), \tag{132}$$

$$\boldsymbol{w}_{t+1} = \boldsymbol{w}_t - k\eta\Big(\frac{1}{kB}\sum_{j<k}\sum_{x\in\mathcal{B}_j}\frac{\partial \boldsymbol{L}}{\partial \boldsymbol{w}}\Big|_{\boldsymbol{w}_t,x} + \lambda\boldsymbol{w}_t\Big). \tag{133}$$

Goyal et al. (2017) shows that combining with gradual warmup, LSP can enlarge the batch size up to $8192(256 \times 32)$ without severe degradation on ImageNet experiments.

LSP has been proven extremely effective in a wide range of applications. However, from the perspective of SMD, the angular update mostly relies on the pre-defined hyper-parameters, and it is hardly affected by batch size. To clarify the connection between LSP and SMD, we explore the learning dynamics of DNN with different batch size by conducting extensive experiments with ResNet50 on ImageNet, the training settings rigorously follow Goyal et al. (2017): momentum coefficient is $\alpha = 10^{-4}$; WD coefficient is $\lambda = 10^{-4}$; Batch size is denoted by $B$; learning rate is initialized as $\frac{B}{256} \cdot 0.1$; Total training epoch is 90 epoch, and learning rate is divided by 10 at 30, 60, 80 epoch respectively.

The results of experiments(Figure 7, 8) suggests that the assumption of LSP does not always hold in practice because of three reasons: first, the approximate equivalence between a single iteration in large batch setting, and multiple iterations in small batch setting can only hold in pure SGD formulation, but momentum method is far more commonly used; Second, according Theorem 2, the enlargement ratio of angular update is only determined by the increase factor of learning rate. Figure 7 shows in practice, the accumulated angular update $\angle(\boldsymbol{w}_t, \boldsymbol{w}_{t+k})$ in small batch batch setting is much larger than angular update $\angle(\boldsymbol{w}_t, \boldsymbol{w}_{t+1})$ of a single iteration in larger batch setting when using Linear Scaling Principle; Third, even in pure SGD cases, the enlargement of angular update still relies on the increase of learning rate, and has no obvious connection to the enlargement of gradient's norm when equilibrium condition is reached (see Figure 8).

In conclusion, though LSP usually works well in practical applications, SMD suggests we can find more sophisticated and reasonable schemes to tune the learning rate when batch size increases.

### C.4    SPHERICAL MOTION DYNAMICS WITH DIFFERENT NETWORK STRUCTURES

We also verify our theory on other commonly used network structures (MobileNet-V2 (Sandler et al., 2018), ShuffleNet-V2+ (Ma et al., 2018)) with standard training settings. The results is shown in Figure 9.

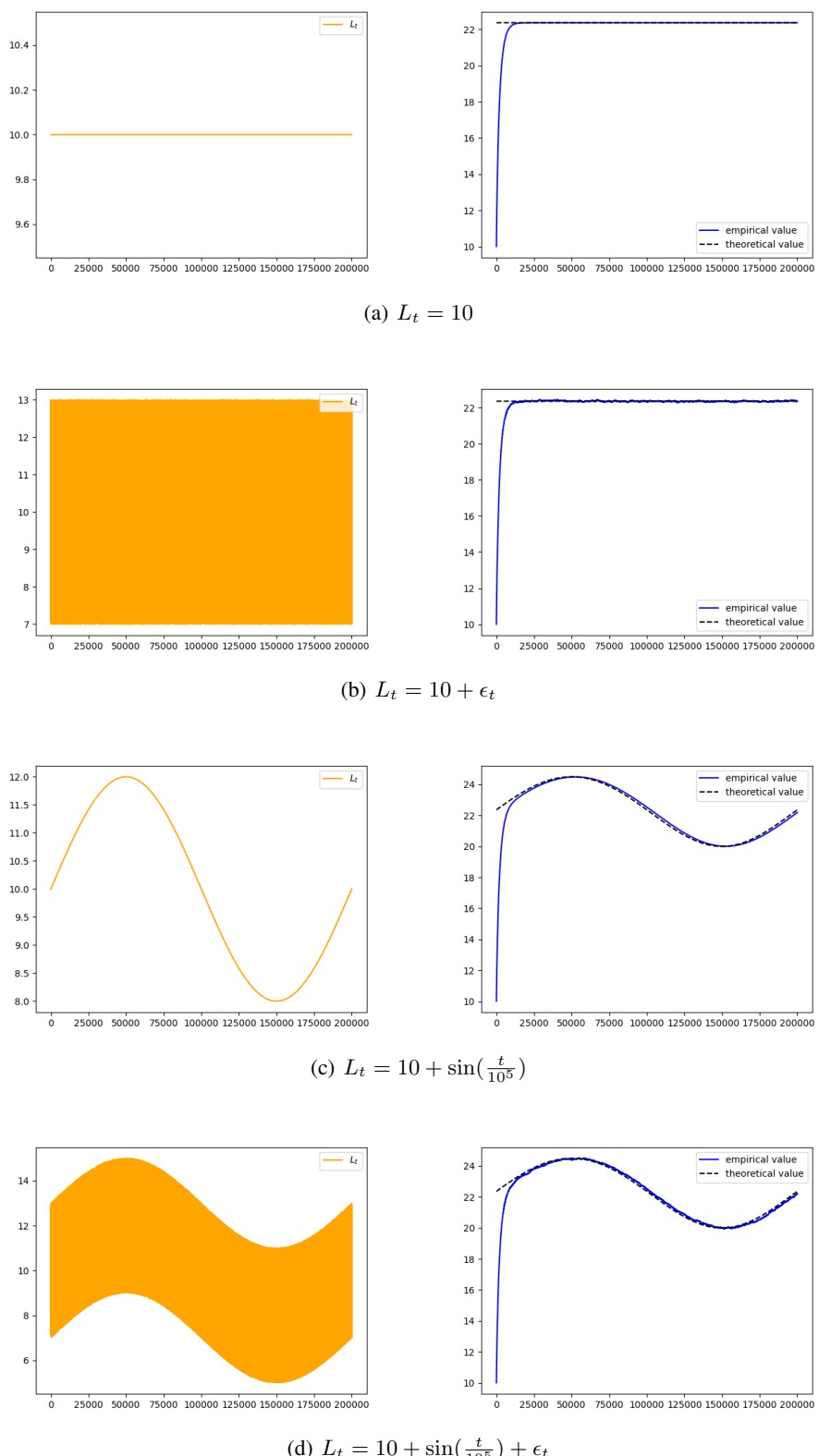

Figure 4: Simulation of SGD (Eq.(130)), $\eta = 0.1$, $\lambda = 0.001$, $x_0 = 10$, $\epsilon_t \sim \mathcal{U}(-3, 3)$. Orange lines represent the square of unit gradient norm; blue solid lines represent simulated value of weight norm square; black dashed lines represent theoretical value of weight norm square

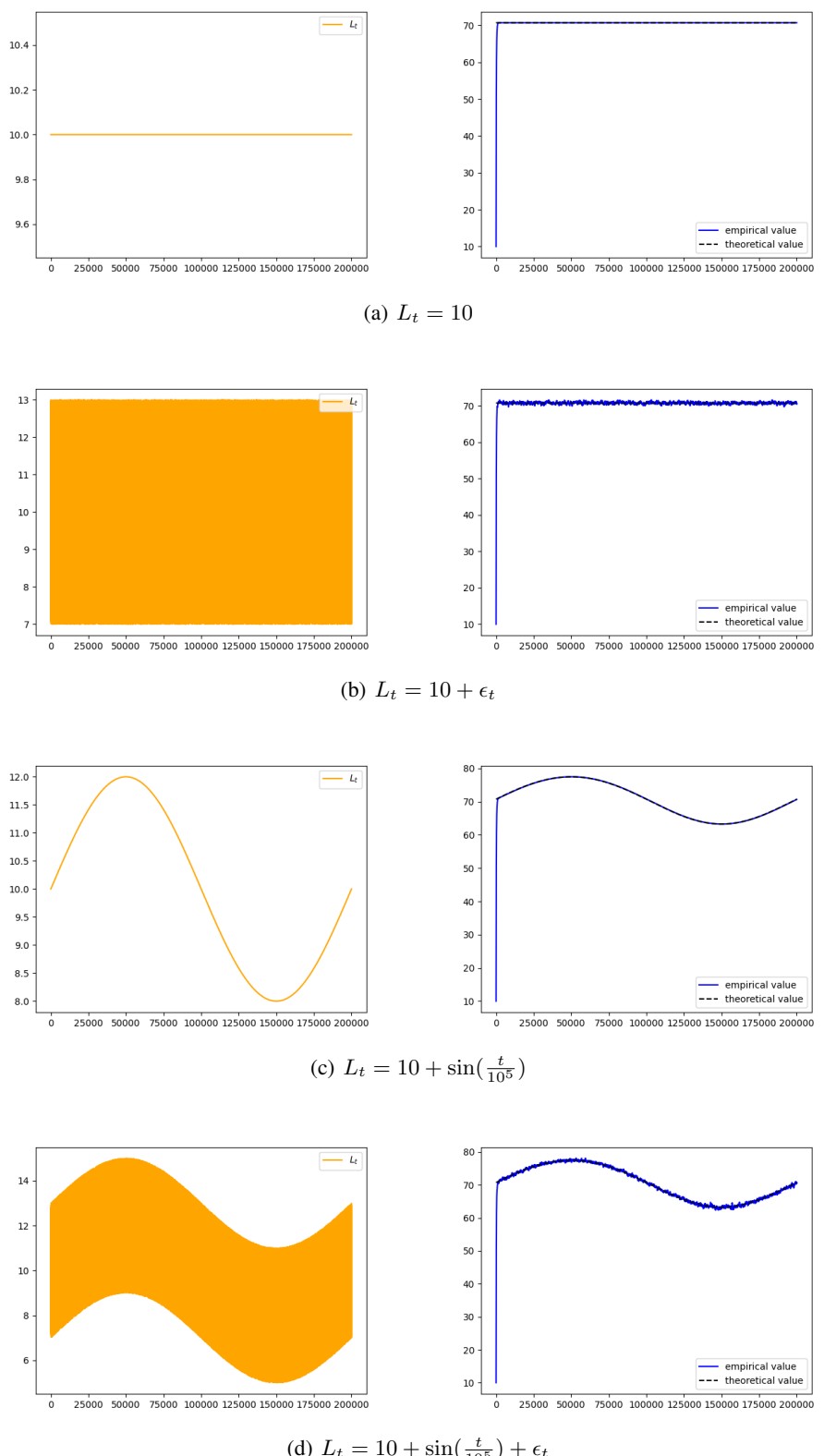

(a) $L_t = 10$

(b) $L_t = 10 + \epsilon_t$

(c) $L_t = 10 + \sin(\frac{t}{10^5})$

(d) $L_t = 10 + \sin(\frac{t}{10^5}) + \epsilon_t$

Figure 5: Simulation of SGDM (Eq.(131)), $\eta = 0.1$, $\lambda = 0.001$, $x_0 = 10$, $\epsilon_t \sim \mathcal{U}(-3, 3)$, $\alpha = 0.9$. Orange lines represent the square of unit gradient norm; blue solid lines represent simulated value of weight norm square; black dashed lines represent theoretical value of weight norm square

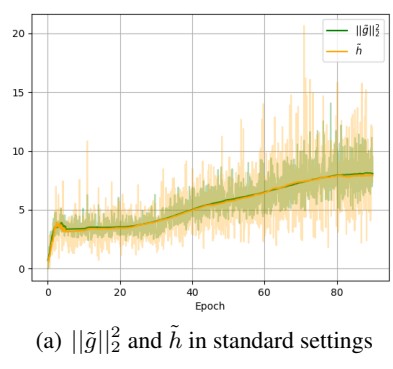

(a) $||\tilde{g}||_2^2$ and $\tilde{h}$ in standard settings

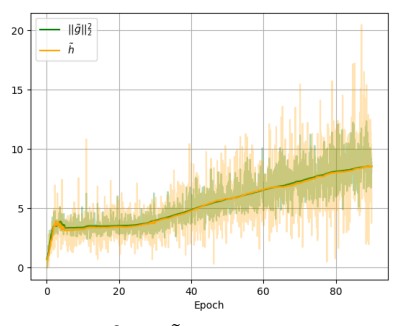

(b) $||\tilde{g}||_2^2$ and $\tilde{h}$ in rescaled settings

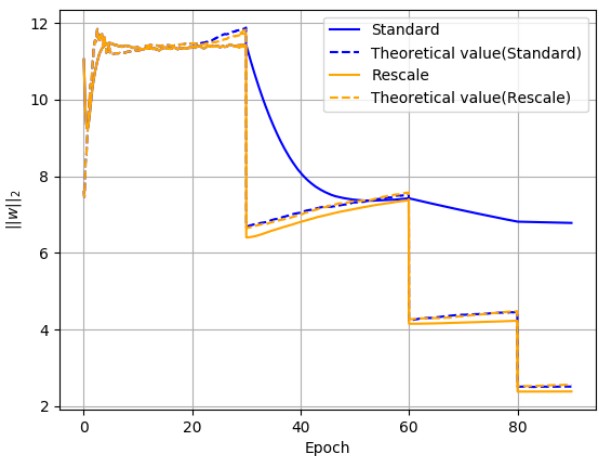

(c) Weight norm in Imagenet with theoretical value

Figure 6: $||\tilde{g}_t||_2^2$, $\tilde{h}_t$ and norm of weight from *layer.1.0.conv2* in Resnet50 backbone. (a),(b) present $||\tilde{g}||_2^2$, $\tilde{h}$ in multi-state learning rate schedule experiments discussed in Section 5.2. semitransparent line represents the raw value of $||\tilde{g}_t||_2^2$ and $\tilde{h}_t$, solid line represents the averaged value within consecutive 200 iterations to estimate the expectations $\mathbb{E}||\tilde{g}_t||_2^2$ and $\mathbb{E}\tilde{h}_t$. (c) presents the empirical value of weight norm and its theoretical value in standard and rescaled cases. The theoretical value is computed by estimated expectations $\mathbb{E}\tilde{h}_t$ in (a), (b) respectively.

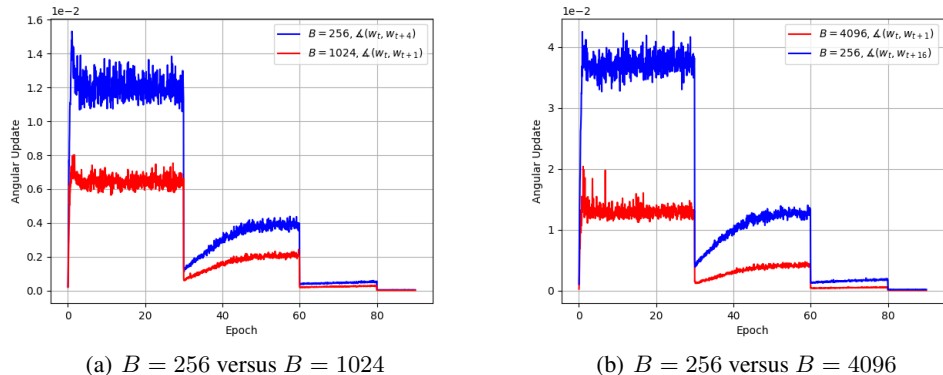

(a) $B = 256$ versus $B = 1024$          (b) $B = 256$ versus $B = 4096$

Figure 7: Angular update of weights from layer1.0.conv2 in ResNet50. The blue lines represent the angular update of weights within a single iteration in when batch settings is $B = 1024(4096)$; The red lines represent the accumulated angular update within 4(16) iterations in smaller batch setting($B = 256$).

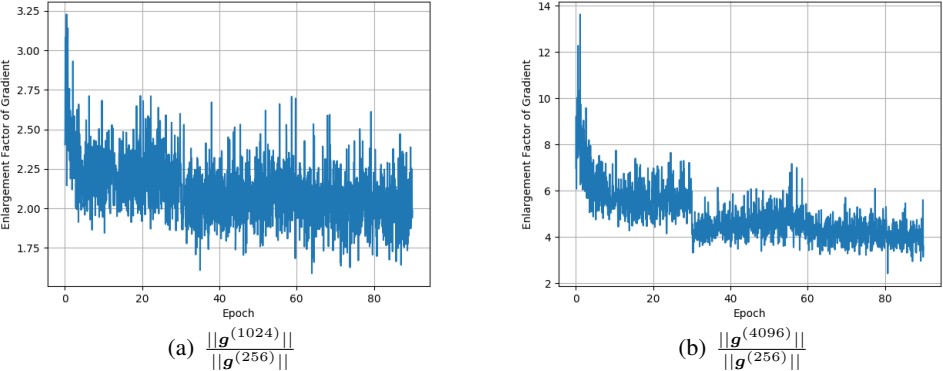

(a) $\dfrac{||\boldsymbol{g}^{(1024)}||}{||\boldsymbol{g}^{(256)}||}$          (b) $\dfrac{||\boldsymbol{g}^{(4096)}||}{||\boldsymbol{g}^{(256)}||}$

Figure 8: Enlargement ratio of gradients' norm of weights from layer1.0.conv2 when batch size increases . $||\boldsymbol{g}^{(k)}||$ represents the gradient's norm computed using $k$ samples(not average).

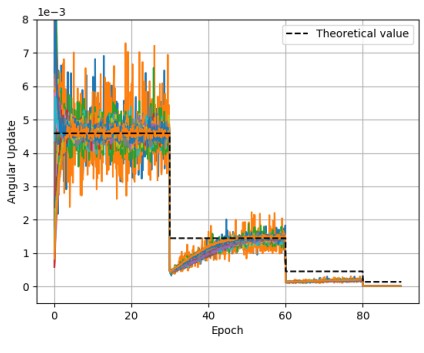

(a) Angle Update of MobileNet-V2

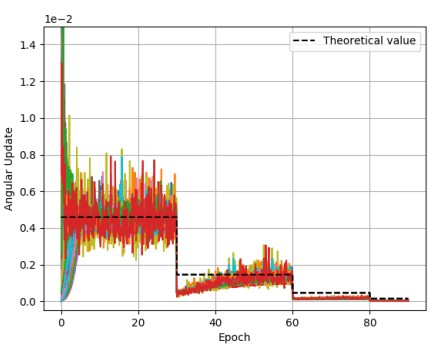

(b) Angular Update of ShuffleNet-V2+

Figure 9: The angular update $\boldsymbol{\Delta}_t$ of MobileNet-V2 (Sandler et al., 2018) and ShuffleNet-V2+ (Ma et al., 2018). The solid lines with different colors represent all scale-invariant weights from the model; The dash black line represents the theoretical value of angular update, which is computed by $\sqrt{\frac{2\lambda\eta}{1+\alpha}}$. Learning rate $\eta$ is initialized as $0.5$, and divided by $10$ at epoch 30, 60, 80 respectively; WD coefficient $\lambda$ is $4 \times 10^{-5}$; Momentum parameter $\alpha$ is set as $0.9$.

