# OpenReview forum: "Spherical Motion Dynamics: Learning Dynamics of Neural Network with Normalization, Weight Decay, and SGD"
_ICLR.cc/2021/Conference — Reject_

### Official Review · AnonReviewer1 · 2020-10-16
**paper is not well-written and the analysis is not rigorous**

**Rating:** 4
**Confidence:** 4

**Review:**

This paper tried to analyze gradient descent with weight decay and momentum for scale-invariant networks. Convergence rate regarding the norm of the iterate is provided.

I think this paper has several weaknesses and hence do not recommend accepting the paper at the current point.

1. The paper is poorly written. The analysis and arguments are very poorly explained.

Page 2:

"If $w_t \approxeq w_{t+1}$, then the weight norm can be approximated as $\|w_t \|_2 \approxeq $..."
--- Why is that?

"while the gradient component provided by WD always tends to reduce weight norm."
--- Why? Is there any proof to support this argument?

The authors say "one can obtain" equality (4). But why (4) holds? it is very unclear and more explanation is necessary

The authors say "one can easily speculate from eq (5) the magnitude of update in SGDM cases should be ..." but the reasoning is unclear.


Page 3:
"Eq.(7) implies gradient norm is influenced by weight norm, but weight norm does not affect the output of DNN"
--- Why?

How to derive (8)?

"If \| w_t \| \approxeq \| w_{t+1} \|_2, one can obtain (10)"
--- What do you exactly mean when you write "\| w_t \| \approxeq \| w_{t+1} \|_2"? and why the condition leads to (10)?

The authors say "one can derive" (12). But again this is unclear.

"From Eq.(7) we can infer that increasing weight norm can lead to smaller gradient norm if unit gradient norm is unchanged."
--- This needs more explanation.

"Arora et al. (2019) proves that full gradient descent can avoid such problem and converge to a stationary point..."
--- To avoid which problem?

"Besides, practical implementation suggests training DNN without WD always suffers from poor generalization"
--- This needs references.

Page 4:

"We can easily derive (15)" How? Also, what does $\approxeq$ on (15) mean? Can you give a precise statement?

"Notice that the core assumption mentioned above is unit gradient norm is unchanged. In fact this assumption can solve the contradiction we present in Section 1: the convergence of weight norm is not equivalent to convergence of weight, steady unit gradient norm can also make weight norm converge"
--- I found the paragraph hard to parse. What do you mean by "unit gradient norm is unchanged"? Of course, convergence of weight norm is not equivalent to convergence of weight. So why is it a "contradiction"?


2. The proof is not rigorous and the writing of the proof is very sloppy.

I believe the proof in the appendix is not rigorous and I feel unease. The authors use the following phrases (not an exhaustive list) to skip a lot of detailed proof, which does not meet the standard of mathematical writing.

"we can easily prove" (page 11)

"we can prove" (page 13)

"can be formulated as" (page 14)

"it's easy to prove" (page 14)

"can be explicitly expressed" (page 14)

"which it's easy to prove"  (page 15)

"it's easy to obtain" (page 16)

"therefore it is easy to derive" (page 17)


3. The theorems are not very insightful.

What insight does the convergence of weight norm provide? for example, it doesn't rule out the case of converging to a bad local min.

What is the meaning of the value $w^* = ( L \eta / 2 \lambda)^{1/4}$? Can you explain why w^* should have the dependency on each parameter?

Does the theorem say that it will converge to the value $w^*$ regardless of any initialized point?

What does the condition $\lambda \eta << 1$ exactly mean?

What does the condition $\| \tilde{g}_t \|^2 > l$ mean? Why does it hold?
When the iterate reaches at a stationary point, isn't it zero?


4. Comparison to the related works are not clear.

Remark 1 and 2 state the results of this paper are "consistent" with some related works but did not provide any elaboration. What results exactly are being compared with? If the results are consistent, then what is new here?
I hope the authors can explain more.

5. Experiment does not really support the theorems.

It seems to me that the authors didn't really show that the weight norm converges to $w^*$ of Theorem 1 or 2.

---

> ### Author Response · Authors · 2020-11-20
> **Response to AnonReviewer1 (1)**
>
> Thanks for your detailed concerns, we will answer it one by one.
>
>  1. "'If $w_t \approx w_{t+1}$, then the weight norm can be approximated as…' --- Why is that?"
>
>  It’s not our claim, it’s from [1]. we don’t think it is reasonable either, so we have discussed its reasonableness in the last paragraph of Sec 3.
>
> 2. "'while the gradient component provided by WD always tends to reduce weight norm.' --- Why? Is there any proof to support this argument?"
>
>  Please see Eqn (14), (15), if you are unfamiliar with weight decay, please refer to [2, 3] for the background of weight decay;
>
> 3. “The authors say 'one can obtain' equality (4). But why (4) holds? it is very unclear and more explanation is necessary”
>
>  Eqn(4) is from [4], you can refer to [4] for more details. Due to the limited length of submission, we do not have enough space to analyze related work with full details.
>
> 4. “The authors say 'one can easily speculate from eq (5) the magnitude of update in SGDM cases should be ...' but the reasoning is unclear.”
>
>  Eqn(5) is from [5], you can refer to [5]. Again, we do not have enough space to analyze related work with full details.
>
> 5. “‘Eq.(7) implies gradient norm is influenced by weight norm, but weight norm does not affect the output of DNN’ --- Why?"
>
>  Eqn(7) shows when weight norm is multiplied by k, gradient norm will be divided by k. Weight norm does not affect the output of DNN because of scale-invariant property.  Please see the definition of scale-invariant property in the second paragraph in Introduction, or refer to related work [1, 4, 5, 6].
>
> 6. "How to derive (8)?“
>
>  By setting $k =1/||w_t||$ in Eqn(7), Eqn (8) is obtained.
>
> 7. “‘If | w_t | \approxeq | w_{t+1} |_2, one can obtain (10)’ --- What do you exactly mean when you write "| w_t | \approxeq | w_{t+1} |_2"? and why the condition leads to (10)?”
>
>  "| w_t | \approxeq | w_{t+1} |_2" means $||w_t||_2$ and $||w_{t+1}||_2$ are very close. Eqn (10) is obtained via dividing Eqn(9) by $||w_t||_2$. This statement comes from [7], you can refer to [7] for more details. We modify the statement in revised version for clarification.
>
> 8. “The authors say "one can derive" (12). But again this is unclear.”
>
>  When $\Delta_t$ is very close to 0, $\tan(\Delta_t) \approx \Delta$ (First order Taylor series expansion). Then combining Eqn. (6), (9), (11), we have Eqn (12).
>
> 9. “‘From Eq.(7) we can infer that increasing weight norm can lead to smaller gradient norm if unit gradient norm is unchanged.’ --- This needs more explanation.”
>
>  You can see Eqn.(8): if unit gradient norm is unchanged, gradient norm is inversely proportional to weight norm. “Inverse proportion” means if weight norm increase, gradient norm will decrease.
>
> 10. “ ‘Arora et al. (2019) proves that full gradient descent can avoid such problem and converge to a stationary point...’ --- To avoid which problem?“
>
>  We present the problem in the sentence exactly before the one you mentioned: “Zhang et al. (2019) states the potential risk that GD/SGD without WD but BN will converge to a stationary point not by reducing loss but by reducing effective learning rate due to increasing weight norm.” We change “problem” to “risk” for clarification in revised version.
>
> [1] Twan van Laarhoven. L2 regularization versus batch and weight normalization. In Advances in Neural Information Processing Systems. 2017.
>
> [2] Alex Krizhevsky and Hinton Geoffrey. Learning multiple layers of features from tiny images. 2009.
>
> [3] Guodong Zhang, Chaoqi Wang, Bowen Xu, and Roger Grosse. Three mechanisms of weight decay regularization. In International Conference on Learning Representations, 2019.
>
> [4] Vitaliy Chiley, Ilya Sharapov, Atli Kosson, Urs Koster, Ryan Reece, Sofia Samaniego de la Fuente, Vishal Subbiah, and Michael James. Online normalization for training neural networks. In Ad- vances in Neural Information Processing Systems 32, pp. 8433–8443. Curran Associates, Inc., 2019.
>
> [5] Zhiyuan Li and Sanjeev Arora. An exponential learning rate schedule for deep learning. In Interna- tional Conference on Learning Representations, 2020.
>
> [6] Sanjeev Arora, Zhiyuan Li, and Kaifeng Lyu. Theoretical analysis of auto rate-tuning by batch normalization. In International Conference on Learning Representations, 2019.
>
> [7] Elad Hoffer, Ron Banner, Itay Golan, and Daniel Soudry. Norm matters: efficient and accurate normalization schemes in deep networks. In S. Bengio, H. Wallach, H. Larochelle, K. Grauman, N. Cesa-Bianchi, and R. Garnett (eds.), Advances in Neural Information Processing Systems 31, pp. 2160–2170. Curran Associates, Inc., 2018.

---

> ### Author Response · Authors · 2020-11-20
> **Response to AnonReviewer1 (2)**
>
> 11. “’Besides, practical implementation suggests training DNN without WD always suffers from poor generalization" --- This needs references.”
>
>  Please refer to [2, 3], we will add reference in revised version. Thanks for pointing out.
>
> 12. “"We can easily derive (15)" How? Also, what does $\approx$ on (15) mean? Can you give a precise statement?”
>
>  $\approx$ in Eqn (15) means we omit a small part $o(\lambda\eta)$. We add the specific derivation in appendix A.1 in revised version. It’s an intuitive analysis to demonstrate the equilibrium condition presented by former work[1,4]. This is not our contribution, we summary it to provide background/preliminary of our topic. The precise statement and rigorous theory, which are our contributions, is presented in the following sections(Sec 4, 5).
>
> 13. “I found the paragraph hard to parse. What do you mean by "unit gradient norm is unchanged"? Of course, convergence of weight norm is not equivalent to convergence of weight. So why is it a "contradiction"?”
>
>  As we stated in the sentence, to understand this paragraph, you might need read the contradiction in [1] we present in the paragraph in left side of Figure 1 in page 2: [1] think equilibrium condition is reached when training converges, that means $w_t = w_{t+1}$, hence we have $||w_t||=||w_{t+1}||$, it can yields $||w_t - w_{t+1}||/||w_{t}|| \approx \sqrt(2\lambda\eta)$, which is contradictory to the condition $ w_t = w_{t+1}$. We solve the "contradiction" by stating “convergence of weight norm is not equivalent to convergence of weight”, it’s our claim to solve the contradictory, not the contradictory we want to clarify. According to Eqn (15), the convergence of weight norm requires unchanged unit gradient norm, that’s why we mention “unit gradient norm is unchanged”. Again, it is just intuitive analysis, rigorous results can be seen in the following sections. We have reorganized the sentences to clarify the misunderstanding in revised version.
>
> 14. “What insight does the convergence of weight norm provide? for example, it doesn't rule out the case of converging to a bad local min.”
>
>  We have added the insight of equilibrium condition in SMD in revised version. This part is placed in the last paragraph of Section 1, remark3, and last paragraph in Section4. As for your example, our theorem implies the cause of equilibrium condition has no connection with the trajectory of optimization or loss landscape. So no matter whether optimization trajectory is trapped in a bad local minimum or has not been close to any local minimum, as long as the assumptions of our theory hold, equilibrium condition always can be reached. So far we cannot have any conclusion if the property of equilibrium condition can help avoid bad local minimum because we do not explore this question in this paper. It is a good question but beyond the discussion of this paper.
>
> 15. “I believe the proof in the appendix is not rigorous and I feel unease. The authors use the following phrases (not an exhaustive list) to skip a lot of detailed proof, which does not meet the standard of mathematical writing.”
>
>  We skip some simple but tedious variable substitution procedure，trying to make reader understand our proof easily, we will refine the proof in the revised version.
>
> 16. "What is the meaning of the value $w^*=(L\eta/^2 \lambda)^{1/4}$? Can you explain why w^* should have the dependency on each parameter?"
>
>  $w^*$ is the theoretical value weight norm should be in equilibrium condition. It means $w^*$ is determined by expectation of square of unit gradient norm ($L$), learning rate ($\eta$), and weight decay factor ($\lambda$). We derive the value of $w^*$ by rigorous mathematical derivation, we think only math knows why $w^*$ should be $w^*=(L\eta/^2 \lambda )^{1/4}$.

---

> ### Author Response · Authors · 2020-11-20
> **Response to AnonReviewer1(3)**
>
> 17. “Does the theorem say that it will converge to the value regardless of any initialized point?”
>
>  Yes, it is, as long as the assumptions always hold. Notice different initialized points may yield different expectation of unit gradient norm square ($E||\tilde{g}||_2^2$), resulting in different value of $w^*$, but the relationship between $w^*$ and $E||\tilde{g}||_2^2$ always hold.
>
> 18. “What does the condition $\lambda\eta << 1$ exactly mean?”
>
>  It means we omit all $o(\eta\lambda)$ part, where $o(\eta\lambda)$ means infinitely small quantity compared with $\lambda\eta$.
>
> 19. “What does the condition $|\tilde{g}_t|^2>l$ mean? Why does it hold? When the iterate reaches at a stationary point, isn't it zero?”
>
>  It means square of unit gradient norm has a lower bound. We verify its reasonableness on Imagenet experiment with standard training settings. Our theory is only applicable when unit gradient norm is not exactly to zero, stationary point case is beyond our topic and discussion. In fact, we never see DNN can meet a stationary point during training procedure, recent publication [8] also points out normalized network may never converge.
>
> 20. “Remark 1 and 2 state the results of this paper are "consistent" with some related works but did not provide any elaboration. What results exactly are being compared with? If the results are consistent, then what is new here? I hope the authors can explain more.”
>
>  In remark 1 and 2, we have specifically point out related work is [1, 4, 5], and we have explicitly present their results in Eqn (4), (5), and “Joint effects of normalization and weight decay.” Paragraph in page 2. We highlight this in revised version. We have stated and highlighted our contribution in introduction: we prove why equilibrium condition can be reached. While relevant work directly assumes equilibrium condition can be reached. Their results fail to explain why equilibrium condition can be reached, or derive the approaching rate. Please read our paper over again.
>
> 21. “It seems to me that the authors didn't really show that the weight norm converges to $w^*$ of Theorem 1 or 2.”
>
>  We compare the theoretical value $w^*$ (black dashed line) and empirical value $||w_t||$(blue solid line) of weight norm in Figure2(b, e). The empirical value agree very well with the theoretical value after early stage of training. Please offer us reasons why you do not believe our experiment results, giving us a chance to defend our results or enhance our justification.
>
> [8] Li, Zhiyuan, Kaifeng Lyu, and Sanjeev Arora. "Reconciling Modern Deep Learning with Traditional Optimization Analyses: The Intrinsic Learning Rate." Advances in Neural Information Processing Systems 33 (2020).

---

### Official Review · AnonReviewer4 · 2020-10-27
**Interesting direction but confusing experiments**

**Rating:** 7
**Confidence:** 4

**Review:**

The main goal of the paper is to establish theoretically some previous known results that for scale invariant networks the weight norm has a fixed point with ||w||^4=eta/lambda ||\tilde{g}|| . They also discuss the angular update, which because of scale invariance is basically equivalent to arccos (1-eta lambda) |w_t|^2/|w_t+1|^2 and it thus comes mainly from the gradient. They have some experiments which they compare with the predicted equilibrium values for the angular update/ weight norm.

The theorems are the main results and I don't think they are very powerful given what is known in the literature (the van Laarhoven paper but also the more recent Li-Arora and Lewkowycz-Gurari where the relevant eta lambda time scale has been shown to be characterize the convergence rate). Also theorem 3 assumes that the normalized gradient norm is constant through training, but it actually changes a lot.

The experiments do not scale invariant networks (even if they have batch-norm, the last layer for example is clearly not scale invariant) so it is not that clear why the theory applies to that case, and the authors should probably discuss this.
In the learning rate decay setup, the experiments don't include the equilibrium value for the weights and it would be good to include them, and it does not seem like the equilibrium analysis is valid there as the authors discuss. The authors propose to rescale the weights together with decaying the learning rate, it would be good if they could mention how this changes the model validation accuracy.

I find it really intriguing that in figures 2ef , the weight norm tracks so well the equilibrium one (which by definition has \Delta |w|=0) but at the same time it, the weight norm changes considerably? Could the authors comment on how this is possible? I suspect that this is because the predicted one is actually really noisy due to batch noise (as shown in figures 2ad) and thus the system is never at equilibrium.

---

> ### Author Response · Authors · 2020-11-20
> **Response to AnonReviewer4 (1)**
>
> Thanks for your comments. We will response your concerns one by one.
>
> 1. “ The theorems are the main results and I don't think they are very powerful given what is known in the literature (the van Laarhoven paper but also the more recent Li-Arora and Lewkowycz-Gurari where the relevant eta lambda time scale has been shown to be characterize the convergence rate)”
>
>  As we emphasize our novelty and contribution in the introduction part, what we really focus on is how equilibrium condition can be reached during the training.  However, all the previous published works before submission due (Oct 2 2020) assume the equilibrium condition has been reached and this strong assumption makes their result impossible to connect with observations in practice. For example, van Laarhoven, 2017 derived the theoretical value of weight norm in equilibrium condition, but his interpretation about equilibrium condition is ambiguous and contradictory, as we discussed in in Sec 1(the paragraph next to figure 1). We solve the contradiction by clarifying the role of unit gradient norm during training. We also derive the theoretical value of weight norm/AU in SGDM case, while results of van Laarhoven 2017 does not include these.
>
>  As for [1] and [2] (we think reviewer may refer to these two papers)，[1] studied the rate of SGD to achieve the best performance of DNN, their analysis has no connection to equilibrium condition of normalized DNN with weight decay. DNN can reach equilibrium condition long before DNN is well trained (get best performance), so [1] is basically irrelevant to our submission.
>
>  [2] indeed explores the cause of equilibrium condition by SDE, but its analysis is limited on pure SGD case and full of conjectures (the discussion about momentum case is only conjecture presented in appendix), and their theory is only qualitative results, unable to derive theoretical value of weight norm/AU. Compared with [2], our work depicts the dynamics of weight norm as an iterative sequence, with mild assumptions that are highly correlated with observations in practice. Our theory on momentum case is the novel result which is rigorously proved, and able to quantify the rate to reach equilibrium condition. We also derive the theoretical value of weight norm/AU in SGDM case. We emphasize that the expansion from SGD case to SGDM case is not trivial. Besides, [2] is a neurlps2020 submission, the paper is revealed to public on Oct 6 2020, 4 days after submission due of ICLR2021, so it is a parallel work.
>
>  In summary, we humbly suggest reviewer can read our paper over again and re-evaluate the novelty/contribution of our work. We have added the comparison with [1] and [2] in revised version.
>
> 2. “Also theorem 3 assumes that the normalized gradient norm is constant through training, but it actually changes a lot.”
>
>  We have modified the statement of theorem 3 to take variance case into account in revised version.
>
> 3. “The experiments do not scale invariant networks (even if they have batch-norm, the last layer for example is clearly not scale invariant) so it is not that clear why the theory applies to that case, and the authors should probably discuss this.”
>
>  We need to emphasize that our theorem focus on scale invariant weights, it is applicable for all scale-invariant weight even they are sub-sets of the whole weights, that is the reason we use partial gradient symbol（$\partial$）not full gradient symbol $\nabla$ in mathematical formulation. The equilibrium condition of each sub-group scale-invariant weights is not affected by other weights’ behavior or performance of networks, including those who are not scale-invariant. This independent property yields the most impressive results: all scale invariant weights from different convolution layers have the same expectation of AU, only determined by hyper-parameters (See figure 3 (a, b, d, e)).
>
> [1] Lewkowycz, Aitor, and Guy Gur-Ari. "On the training dynamics of deep networks with $ L_2 $ regularization." Advances in Neural Information Processing Systems 33 (2020).
>
> [2] Li, Zhiyuan, Kaifeng Lyu, and Sanjeev Arora. "Reconciling Modern Deep Learning with Traditional Optimization Analyses: The Intrinsic Learning Rate." Advances in Neural Information Processing Systems 33 (2020).

---

> ### Author Response · Authors · 2020-11-20
> **Response to AnonReviewer4 (2)**
>
> 4. “In the learning rate decay setup, the experiments don't include the equilibrium value for the weights and it would be good to include them, and it does not seem like the equilibrium analysis is valid there as the authors discuss.”
>
>  We have to point out that regarding equilibrium condition as a static state, where weight norm remains unchanged forever, is a stereotype caused by ambiguous discussion in previous important literature [3,4]. Our theory shows equilibrium condition is actually a dynamic condition, it can be rapidly reached and continue with fixed hyper-parameters and slowly changing $E||\tilde{g}||_2^2$, but it will be broken when learning rate decay abruptly and significantly, new equilibrium condition need to be reached after that. So the whole contents in Sec 5.2 demonstrates the process from old equilibrium condition to the new one, by both theoretical analysis (Corollary 3.1.) and experiments (Figure 3). We hope reviewer 3 can get rid of stereotype of equilibrium condition, and read Sec 5.2 again to get our point.
>
>  We have added theoretical value of weight norm in learning rate decay experiments, and present the performance of neural network in un-rescaled/rescaled cases(see Figure 6, table 1, 2 in appendix), but we need to emphasize: we design rescaled strategy only for justifying our analysis on the behavior of AU in learning rate decay cases, not for proposing new method to improve the performance of neural network.
>
> 5. “I find it really intriguing that in figures 2ef , the weight norm tracks so well the equilibrium one (which by definition has \Delta |w|=0) but at the same time it, the weight norm changes considerably? Could the authors comment on how this is possible? I suspect that this is because the predicted one is actually really noisy due to batch noise (as shown in figures 2ad) and thus the system is never at equilibrium.”
>
>   We have interpreted the reason (why though unit gradient norm varies a lot across the whole training process，it still can make weight norm track well with its theoretical value) in the paragraph below Remark 1, we also conduct experiments on synthetic data to verify our analysis (see Appendix B.1 (C.1 in revised version)). To make this point crystal clear, here we give a further explanation.
>
>  “$||w||_2$ remains unchanged ($\Delta||w||_2=0$) in equilibrium condition” is a stereotype our paper intends to dispel. the change of $||\tilde{g}_t||_2^2$ can be divided into two parts: local variance($Var(||\tilde{g}_t||_2^2)$) and change of expectation ($E||\tilde{g}_t||_2^2$). Our theorems show local variance ($V$) will make $||w_t||_2^2$ noisy even in equilibrium condition, but $var(||w_t||_2^2)$ is so small (bounded by $V\eta^2/l$) that $||w_t||_2^2$ seemingly unchanged; As for change of $E||\tilde{g}||_2^2$, the change is too slow: in Figure 1(a), we present records of the whole 225,000 steps. If we observe the change of $E||\tilde{g}||_2^2$ within 1,000 iterations, $E||\tilde{g}||_2^2$ is nearly constant. Slow change of $E||\tilde{g}||_2^2$ and fast approaching rate of $||w_t||_2^2$ can make $||w_t||_2^2$ always get close to its theoretical value in fixed learning rate case. Therefore, the suspicion in the comment is wrong: the real weight norm is actually the noisy one (solid blue line in Figure 2(b, e), note it is a raw record, we do not smoothen it), but its variance is so small that make it seemingly smooth; the predicted line is computed by $E||\tilde{g}_t||_2^2$ or $E\tilde{h}_t$ estimated in Figure 2(a, d)(solid line), since $E||\tilde{g}_t||_2^2$ or $E\tilde{h}_t$ changes smoothly, the predicted line of weight norm is smooth too. Additionally, to verify these arguments, please see the proof of theorem 1,2 in Sec B in appendix, and the experiments on synthetic data in Sec C.1 in appendix to see how noisy unit gradient norm can yield (seemingly) smoothly changing weight norm.
>
> [3] Twan van Laarhoven. L2 regularization versus batch and weight normalization. In Advances in Neural Information Processing Systems. 2017.
>
> [4] Vitaliy Chiley, Ilya Sharapov, Atli Kosson, Urs Koster, Ryan Reece, Sofia Samaniego de la Fuente, Vishal Subbiah, and Michael James. Online normalization for training neural networks. In Ad- vances in Neural Information Processing Systems 32, pp. 8433–8443. Curran Associates, Inc., 2019.

---

> > ### Comment · AnonReviewer4 · 2020-11-23
> > **Response**
> >
> > Thanks for the detailed comments. I understand your perspective and none of my points referred to the "stereotypical" point of view. Thanks for adding the performance tables and the "nonstereotypical" equilibrium comparison for point [4].
> >
> > I still don't understand the content of this non-stereotypical equilbrium if this does not slow down the growth of the weight norm. The derivation of the theoretical curve comes from solving $\Delta |w|^2=0$ as a function of $|\tilde{g}|^2$, of course since this is not updated instantly and $|\tilde{g}|$ changes, this $\Delta |w|^2$ will be small, however ,$\Delta |w|^2$ seems to be the same order whether it is on equilibrium or not (like in 2.e), I guess it must be because $\tilde{g}|^2$ is larger close to dynamical equilibrium. My question is, is the weight norm curve exhibiting any characteristic behaviour (ie characterized solely by its curve and
> > its derivatives) when it is in this dynamical equilibrium?

---

> > > ### Author Response · Authors · 2020-11-24
> > > **Thanks for your quick reply**
> > >
> > > Very thanks for your quick reply. As for your question, according to our theorem, the weight norm curve in equilibrium indeed has no intrinsic characteristic, because 1) the norm of scale-invariant weight do not affect the performance of DNN at all, unless it is exactly to $0$; and 2)  the norm of weight in equilibrium condition is totally determined by expected unit gradient norm square and pre-defined hyper-parameters. Unit gradient norm is an intrinsic indicator of DNN, which is not affected by the scale-invariant property.
> > >
> > > However, our results do not deny the effectiveness of techniques which are based on scale of weight norm (like pruning methods[1, 2]). Though weight norm has no intrinsic characteristic, it can reflect the scale of expected unit gradient norm square ($w^* \propto (E||\tilde{g}||_2^2)^{1/4}$).  So we think weight norm in equilibrium is meaningful to reflect some state of DNN by revealing information of unit gradient norm.
> > >
> > > In summary, we think behavior of weight norm only matters in interpreting the cause of equilibrium condition. When equilibrium condition has been reached, behavior of unit gradient norm and hyper-parameters (learning rate, weight decay factor, momentum factor) dominate all.
> > >
> > > [1] Li H, Kadav A, Durdanovic I, et al. Pruning filters for efficient convnets, ICLR, 2017.
> > >
> > > [2] Liu Z, Li J, Shen Z, et al. Learning efficient convolutional networks through network slimming[C]//Proceedings of the IEEE International Conference on Computer Vision. 2017: 2736-2744.

---

> > > > ### Comment · AnonReviewer4 · 2020-11-25
> > > > **Upgraded rating**
> > > >
> > > > After reading the authors responses and thinking more carefully about the paper, I have upgraded the rating of the paper and would like to see it accepted. I think the phenomena the authors are describing, the "dynamical equilibration" of the weight norm and the constant angle between weights (only dependent on hyperparameters) are key features that can help understand better the optimization of (mostly) scale invariant networks. These two features are robustly confirmed by the experiments and the derivation of the equilibrium values is very simple.

---

### Official Review · AnonReviewer3 · 2020-10-28
**Some Comments**

**Rating:** 5
**Confidence:** 3

**Review:**

The equilibrium condition is important during the theoretical analysis of spherical motion dynamics.
This paper focuses on the reasonableness of the equilibrium condition.
Concretely, the authors first show that the weight norm can converge to its theoretical result under some assumptions (with SGD, SGDM settings).
Finally, the experiments verify their conclusions.

### CONTRIBUTIONS

a) The analysis of the equilibrium condition is important. The authors analyzed the equilibrium conditions under different settings, including SGD, SGDM, angular.

b) The problem is motivated well. The authors gave a detailed introduction to the question of why we need to analyze the equilibrium conditions.

c) The authors conducted several experiments to show the reasonableness of the theorems/assumptions. And the theoretical findings agree well with empirical observations.

### MAJOR CONCERNS

I am still concerned with the explanations in Eqn.17. (Of course, the same concern in Eqn.20.)
In Eqn.17, the formula (left) is the MSE form, which is the combination of the bias and the variance.
The authors explained it like “the square of weight norm can linearly converge to its theoretical value in equilibrium, and its variance is bounded by the variance of the square of unit gradient norm multiplying the square of learning rate”.
Could the authors tell us why the first term (vanish as t \to \infty) on the right hand is actually the bias?
In my view, one cannot say the estimator *converge* to its theoretical value if the right-hand does not converge to zero.

I think the most important point in the paper is the convergence of the estimator to the theoretical value, which is also the thing the authors tried to verify in Figure2 (b, e).
To reach the conclusion, one needs to at least claim “V\eta^2/l” is small. As shown in Figure2 (a), I do not think V is such small.
Also, the claims in the main text that “the learning rate \eta is small” is not satisfying. I am not sure if I have skipped some important claims in the main text.
I would like to raise my score if the authors can explain it better.

### OTHER CONCERNS

a) Eqn.12 (as well as its statements) is not clear. It is still unclear to me why one needs to use an angular update.

b) There are several typos (and also something like log->\log, min->\min) in the appendix. I believe the conclusions are mostly correct (although I did not check the details), but I still spend a hard time reading the appendix.

c) In Theorem3, the assumption “when V=0” is a little bit strong and unrealistic (shown in Figure2(a)). The importance of the theorem will be larger if the authors could use a mild assumption.

---

> ### Author Response · Authors · 2020-11-20
> **Response to AnonReviewer3**
>
> We appreciate your valuable comments. We will response your concerns one by one.
>
> Major concerns:
>  1. "…one cannot say the estimator converge to its theoretical value if the right-hand does not converge to zero."
> We want show that MSE between $||w_t||_2^2$ and $(w^*)^2$ can approach to a small value $V\eta^2/l$ in a linear rate regime, when the vanishing term $(1-4\lambda\eta)^TB$ is larger than the variance term $V\eta^2/l$. We will correct the misleading statements in the revised version.
>  2. "…I do not think V is such small. Also, the claims in the main text that “the learning rate \eta is small” is not satisfying…”
> We do not state $V\eta^2/l$ is small due to small value of $V$, we need to clarify that the denominator $l$ ($l$ denotes the lower bounder of square of unit gradient norm) is not small either, and $\eta$ is set as 0.1 in our experiment. For example, in Figure 2(a) (semi-transparent line does not represent standard deviation bar, but exact experiment records.), at epoch 40, $V<4, l>6, \eta=0.2$, then $V\eta^2/l < 0.027$,. According to Figure 2(b), $||w_t||_2^2 > 64$, therefore $ V\eta^2/l << ||w_t||_2^2$, our experiment results strongly verify our claim that $V\eta^2/l$ is small. We will refine the statement to emphasize this point in revised version.
>
> Other concerns:
>
> a)	Eqn.12 wants to show the connection between angular update (AU) and ELF defined in [1](Eqn.(3), we will refine the statement for clarification. We propose AU for two reasons: first, we intend to replace the ambiguous definition “effective learning rate (ELF)”with AU to depict the change of scale-invariant weight. Previous important literatures ([1], [2], [3]) give different meanings to ELF, because they choose different forms of gradient norm to multiply; Second, AU is the intrinsic meaning of relative update $||w_{t+1} – w_t||/||w_t||$ within equilibrium condition, which has been mentioned in previous works ([1], [2], [3]). $ w_{t+1} – w_t $ and $w_t$ are both vectors, and $w_t$ is a scale-invariant weight, so $||w_{t+1} – w_t||/||w_t||$ cannot truly indicate the extent of changes if the equilibrium condition has not been reached yet, while AU can represent the change of $w_t$ well at any time during training process.
>
> Moreover, AU is highly correlated to our most impressive results (see Figure 3 (a, b, d, e)): all scale-invariant (sub)weights in convolutional layers has the same (expected) AU in equilibrium condition, no matter which layer they belong to, what size (number of parameters) they have, what shape they have, or how large their gradient/unit gradient norm is. Both our theory and experiment results imply expected AU of all scale invariant weight is only determined by hyper-parameters. This property of AU is amazing, but it is caused by standard SGD/SGDM, we believe it has great potential to explore the behavior of AU when analyzing learning dynamic of normalized network.
>
> b)   We will check the proof over again and correct the typos, the proof of our theory is not easy and straightforward, we will try our best to make it more readable.
>
> c).  We have modified the statement of theorem 3 to take variance case into account in revised version.
>
> [1]  Twan van Laarhoven. L2 regularization versus batch and weight normalization. In Advances in Neural Information Processing Systems. 2017.

---

### Official Review · AnonReviewer2 · 2020-10-28
**Comments**

**Rating:** 6
**Confidence:** 2

**Review:**

Unfortunately, I am not qualified enough in the literature of learning dynamics of the neural network to comment on the novelty and contribution of this paper. Therefore I may take the author’s word and would like to rely on the comment on other reviewers.

In general, the motivation of this paper is clear, since the theoretical result  in literature mainly relies on the assumption of equilibrium condition but does not discuss why this equilibrium can be reached.

The main contribution comes from  Theorem 1 and Theorem 2.  This paper proves that weight norm can converge at the linear rate under mild assumption . It also defines another index called angular update to measure the change of normalized neural network in a single iteration and prove its convergence with linear rate. Overall the theoretical result looks novel and the experimental result matches the theory.Hence, I lean toward accepting.

My only concern is whether this theory can help us to train the deep neural network or give us some insights to design certain layers of the neural network. But it maybe that the overall novelty/contribution outweights this concerns. Can the author further explain that?

---

> ### Author Response · Authors · 2020-11-20
> **Response to AnonReviewer2**
>
> Thanks for your recognition on our work. Due to the limited length of submission, we cannot discuss too much on the significance of our theory in the paper.
> Our theory is highly correlated with large batch training issue. A typical strategy to deal with large batch training issue is Linear Scaling Rule [1] (see further discussion in appendix B.2(C.3 in revised version)), our theory can show what will happen if we apply Linear Scaling Rule, and can explain why Linear Scaling Rule will fail with extremely large batch size. Therefore, our theory has the potential to inspire new and effective learning rate tuning schedule; Another popular large batch training method LARS [2], is also highly correlated with our theory: the intuition of LARS is to make the relative update of weights in each layer, which happens to coincide with the performance of angular update in SGD/SGDM. But LARS is very sensitive, and its performance is very hard to reproduce, which prevents it from being widely used as SGD or Adam. We believe our theory has the potential to help fix its weakness and make it as stable as SGDM/Adam in large batch training issue.
>
> [1] Priya Goyal, Piotr Dolla ́r, Ross Girshick, Pieter Noordhuis, Lukasz Wesolowski, Aapo Kyrola, An- drew Tulloch, Yangqing Jia, and Kaiming He. Accurate, large minibatch sgd: Training imagenet in 1 hour. arXiv preprint arXiv:1706.02677, 2017.
> [2] You, Yang, et al. "Imagenet training in minutes." Proceedings of the 47th International Conference on Parallel Processing. 2018.

---

### Author Response · Authors · 2020-11-24
**Revised version has been uploaded.**

Dear all,

We have uploaded the revised version of our submission. Following the advice of reviewers, we mainly did following modifications in revised version:

1. Modified theorem 3 to take noisy case into account;

2. Added complementary results (multi-stage case in Sec 5.2) in appendix;

3. Added more discussion about insight of our theorem in main text;

4. Added more comparison with relevant work;

5. Correct misleading statements and typos in main text and appendix.

Best wishes,

Authors of Paper510

---

### Comment · ~Daniel_LK_Yamins1 · 2021-01-20
**This is an important result**

As a reader I strongly feel the result of this paper is important and useful.   I think it is rather too bad that it did not go through  on this review.

First of all, the results are actually correct -- we re-derived the main results of this paper using other methods.  (see https://arxiv.org/abs/2012.04728).    We have also reproduced the authors' experiments showing that the results are empirically correct.

Secondly, they were quite novel: they say the limiting trajectory of the gradient descent for a class of network architectures that includes real-world examples at scale,  moves at a fixed speed on a fixed-norm sphere.  That's very non-trivial.

I hope these results get more expose at some point soon.

---

> ### Comment · ~Ruosi_Wan4 · 2021-01-20
> **Thanks for your support**
>
> Dear Daniel,
>
> We really appreciate your support, your recognition means a lot to us. We will try our best to refine our paper, and make it recognized by the community.
>
> By the way, I've read your paper on arxiv before, I'm really impressed you derive the precise prediction of weight norm using time-continuous theoretical tools, I personally used to think it is impossible to establish precise quantitative results on spherical dynamics and equilibrium in time-continuous settings. Maybe we can discuss on this topic further.
>
> Best,
> Ruosi Wan

---

### Decision · Program_Chairs · 2021-01-07
**Final Decision**

**Decision:**

Reject

**Comment:**

This paper studies gradient descent with weight decay and momentum for scale-invariant networks. While this is an interesting research direction, the clarity of the paper suffers from poor writing. In its current form, I doubt this paper would have a large impact in the community. In addition, some reviewers pointed out that the analysis is not rigorous (some steps are missing, the authors re-use some unjustified steps from previous papers, ...). The authors claim these steps are obvious, I certainly don't think that's the case and at the very least, one would expect that detailed and rigorous derivations in the appendix if the lack of space is the real issue in the main paper.

I think the feedback provided by the reviewers is valuable and I strongly encourage the authors to take advantage of this feedback to improve their work and submit to another venue.